# Hi-SAFE: Hierarchical Secure Aggregation for Lightweight Federated Learning

## Abstract

Federated learning (FL) faces challenges in ensuring both privacy and communication efficiency, particularly in resource-constrained environments such as Internet of Things (IoT) and edge networks. While sign-based methods, such as sign stochastic gradient descent with majority voting (SIGNSGD-MV), offer substantial bandwidth savings, they remain vulnerable to inference attacks due to exposure of gradient signs. Existing secure aggregation techniques are either incompatible with sign-based methods or incur prohibitive overhead. To address these limitations, we propose *Hi-SAFE*, a lightweight and cryptographically secure aggregation framework for sign-based FL. Our core contribution is the construction of efficient majority vote polynomials for SIGNSGD-MV, derived from Fermat's Little Theorem. This formulation represents the majority vote as a low-degree polynomial over a finite field, enabling secure evaluation that hides intermediate values and reveals only the final result. We further introduce a hierarchical subgrouping strategy that ensures constant multiplicative depth and bounded per-user complexity, independent of the number of users $n$. Hi-SAFE reduces per-user communication by over 94% when $n \geq 24$, and total cost by up to 52% at $n = 24$, while preserving model accuracy. Experiments on benchmark datasets confirm the scalability, robustness, and practicality of Hi-SAFE in bandwidth-constrained FL deployments.

## 1 Introduction

Federated learning (FL) facilitates collaborative model training across decentralized clients without exposing raw data (McMahan et al., 2017; Li et al., 2020; Hong & Chae, 2021; Kwon et al., 2023; Lim et al., 2020; Yang et al., 2023), offering intrinsic privacy benefits that make it particularly attractive for sensitive domains such as healthcare, finance, and the Internet of Things (IoT). Nonetheless, deploying FL on real-world edge or IoT devices introduces significant challenges due to limited communication bandwidth, computational capacity, and vulnerability to privacy leakage through shared model updates (Lyu et al., 2022; Nguyen et al., 2021; Aledhari et al., 2022; Kairouz et al., 2021). Although FL effectively preserves data locality, numerous studies have shown that intermediate model updates, such as gradients, can be exploited by adversaries to reconstruct sensitive inputs or perform membership inference (Zhu et al., 2019; Hitaj et al., 2017; Geiping et al., 2020; Nasr et al., 2019; Wei & Liu, 2021). These threats are especially pronounced in resource-constrained environments where devices continuously collect and transmit private information.

To address this, various secure aggregation methods have been proposed. Pairwise additive masking (Bonawitz et al., 2017; So et al., 2022) protects individual updates via secret sharing but may still expose intermediate aggregation results under semi-honest assumptions. Differential privacy (DP) (Truex et al., 2019; Lyu, 2021) provides formal privacy guarantees but often compromises model accuracy due to added noise. Homomorphic encryption (HE) (Cheon et al., 2017; Fang & Qian, 2021) provides strong cryptographic guarantees by enabling computations directly on encrypted data without decryption. However, this approach entails substantial computational and communication costs, which significantly limits its practicality in resource-constrained edge devices.

In parallel, sign-based methods such as SIGNSGD and its majority vote variant SIGNSGD-MV (Seide et al., 2014; Bernstein et al., 2018a;b; Park & Lee, 2023; Jin et al., 2024; Joo et al., 2025) provide exceptional communication efficiency by quantizing updates to 1 bit per parameter.

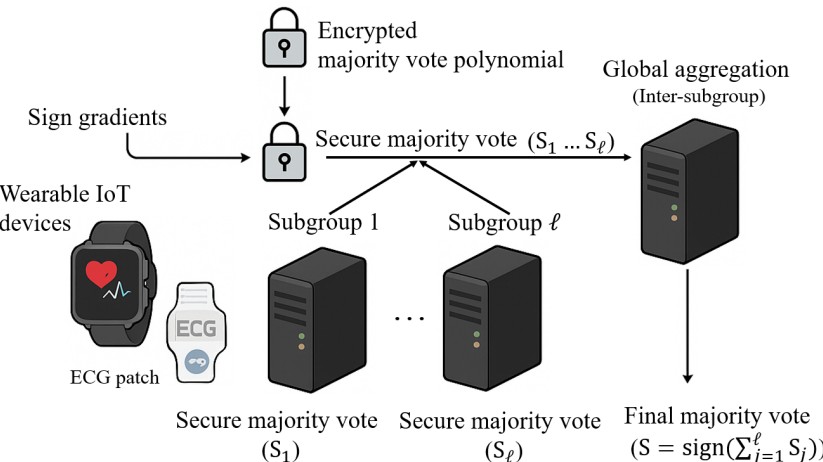

Figure 1: Overview of **Hi-SAFE**. Sign-based local gradients are securely aggregated in a two-level hierarchy through polynomial evaluation. Each user contributes its 1-bit gradient for privacy-preserving intra-subgroup aggregation, and only the final majority vote is revealed to the server, ensuring end-to-end privacy without exposing individual inputs.

These methods are both scalable and robust to noise; however, they expose raw sign gradients to the server, rendering them susceptible to inference attacks (Geiping et al., 2020). Moreover, most existing secure aggregation protocols are either inefficient or fundamentally incompatible with sign-based methods. Specifically, masking-based approaches permit the server to access intermediate summation values during the computation of the final majority vote, which may lead to information leakage. HE cannot directly support nonlinear operations—such as the sign function and majority voting—required by SIGNSGD-MV. Additionally, the large ciphertext sizes in HE undermine the key benefit of 1-bit update protocols. A comprehensive comparative summary of these approaches is presented in Appendix B, highlighting their limitations in the context of sign-based FL.

*These limitations motivate the necessity for a novel class of secure aggregation frameworks that not only preserve the communication efficiency characteristic of sign-based methods but also provide strong privacy guarantees.*

**Contributions.** To address this challenge, we propose *Hi-SAFE* (**Hi**erarchical **S**ecure **A**ggregation for **FE**derated Learning)—a lightweight and cryptographically secure aggregation framework tailored to SIGNSGD-MV. Hi-SAFE minimizes communication cost, protects against inference attacks by revealing only the final majority vote, and scales efficiently in resource-constrained environments. As illustrated in Figure 1, each user contributes a 1-bit signed update that is securely processed through evaluation of majority vote polynomial in a hierarchical structure. Our main contributions are summarized as follows:

- **Cryptographic Secure Aggregation:** We design a privacy-preserving protocol for sign-based FL that discloses only the final majority vote to the server, thereby ensuring protection against inference attacks under the semi-honest model. To the best of our knowledge, this is the first work to provide end-to-end privacy within sign-based FL frameworks.

- **Efficient Majority Vote Polynomial:** Based on Fermat's Little Theorem, we construct the majority vote as a low-degree polynomial over a finite field and show that its secure evaluation is equivalent to the standard (non-private) SIGNSGD-MV, guaranteeing both correctness and privacy.

- **Hierarchical Scalability:** We introduce a subgrouping strategy that maintains constant multiplicative depth (about two subrounds) and a bounded secure multiplication cost ($\leq 6$ per user), independent of the total number of users $n$.

- **Communication-Efficient and Robust Framework:** Hi-SAFE reduces per-user communication costs by over 94% when $n \geq 24$, and achieves up to 52% reduction in total communication at $n = 24$ compared with the non-subgrouping, while preserving or even improving model accuracy. Extensive experiments on benchmark datasets confirm its scalability, robustness, and practicality in bandwidth-constrained FL deployments.

## 2 HI-SAFE DESIGN

### 2.1 PROBLEM SETTING AND DESIGN CRITERIA

We design Hi-SAFE under the following FL environment. We adopt the semi-honest model, in which all users comply with the protocol, although some may attempt to infer private information from the exchanged messages (Bonawitz et al., 2017; Zhang et al., 2023; Zhao, 2023; Jiang et al., 2024; Liu et al., 2024). In addition, we employ the SIGNSGD-MV update rule, whereby each user transmits only the 1-bit sign of its local gradient, and the server determines the global update direction by performing a majority vote over all received signs (Seide et al., 2014; Bernstein et al., 2018a;b; Park & Lee, 2023; Jin et al., 2024).

Based on this setting, Hi-SAFE is designed to achieve both communication efficiency and strong cryptographic privacy through the following core components.

1. **Majority Vote Polynomial $F(\mathbf{x})$ (see Section 2.2.1):** Based on Fermat's Little Theorem, we propose a finite field polynomial $F(\mathbf{x})$ that performs majority voting over sign gradients. This polynomial reproduces the standard majority vote result directly, without the need to compute any intermediate values.

2. **Secure Polynomial Evaluation (see Section 2.2.2):** Each user securely evaluates its additive secret share of the polynomial $F(\mathbf{x})$ without revealing its input $\mathbf{x}$. In this work, we adopt Beaver triples (Beaver, 1991) for secure multiplication, which mask the user's input and yield encrypted shares for aggregation; however, other secure multiplication techniques (e.g., DN (Damgård & Nielsen, 2007) and ATLAS (Goyal et al., 2021)) can be seamlessly integrated into our framework.

3. **Secure Aggregation and Broadcasting (see Section 2.3):** The server aggregates the encrypted shares by summation to compute the final majority vote $F(\mathbf{x})$, which is then broadcast to users for model update. Only the final result is revealed; all intermediate values remain hidden.

4. **Hierarchical Aggregation via Subgrouping (see Section 2.4):** To mitigate the overhead associated with secure polynomial evaluation using techniques such as Beaver triples, which grows significantly as the number of users increases, we divide users into subgroups that perform independent intra-subgroup aggregation. The final result is then obtained by aggregating the outputs of each subgroup, enabling both scalability and privacy while keeping the computational and communication costs manageable.

### 2.2 SECURE EVALUATION OF MAJORITY VOTE POLYNOMIAL $F(\mathbf{x})$ OVER $\mathbb{F}_p$

This section presents how to construct and securely evaluate the majority vote polynomial $F(\mathbf{x})$ over $\mathbb{F}_p$ in an FL setting where $\mathbb{F}_p$ is a prime field for a prime $p$. The primary goal is to compute majority votes over sign gradients while preserving each user's input privacy under an honest-but-curious setting. Appendix C presents an illustrative example for further clarification.

### 2.2.1 MAJORITY VOTE POLYNOMIAL CONSTRUCTION VIA FERMAT'S LITTLE THEOREM

Fermat's Little Theorem enables the construction of an indicator polynomial over $\mathbb{F}_p$, which evaluates to 1 if the input equals a target value and 0 otherwise (Smith, 2020). Building upon this principle, we define the majority vote polynomial $F(\mathbf{x}) = \text{sign}(\mathbf{x})$, where $\mathbf{x} = \sum_{i=1}^{n} \mathbf{x}_i \in \mathbb{F}_p$ for $\mathbf{x}_i \in \{-1, +1\}^d$, and $d$ denotes the vector dimension. Let $p$ be the smallest prime greater than $n$. The challenge lies in expressing this discrete decision function as a finite field polynomial.

When $n$ is even, a tie ($\mathbf{x} = 0$) may occur. Two common tie-breaking rules are:

- $\text{sign}(0) \in \{-1, +1\}$: tie resolved to binary decision (1-bit output of $F(\mathbf{x})$),
- $\text{sign}(0) = 0$: tie represented as a distinct third state (2-bit output of $F(\mathbf{x})$).

This choice affects both the structure of $F(\mathbf{x})$ and the required communication bandwidth. We propose the majority vote polynomial with $d$-dimensional; its $i$-th component is defined as:

$$F(\mathbf{x}) = \sum_{\mathbf{m} \in \{-n, -n+2, \ldots, n-2, n\}} \text{sign}(\mathbf{m}) \cdot \left[ 1 - (\mathbf{x} - \mathbf{m})^{p-1} \right] \pmod{p}, \quad (1)$$

where $\mathbf{m} = \sum_{i=1}^{n} \mathbf{m}_i$ with $\mathbf{m}_i \in \{-1, +1\}$ and $\text{sign}(0)$ is defined by the tie-breaking policy.

**Lemma 1** (Correctness of the Majority Vote Polynomial). *Let* $\mathbf{x}_i \in \{-1, +1\}^d$ *for all* $i \in [n] :=$ $\{1, 2, ..., n\}$, *and define the aggregated value* $\mathbf{x} = \sum_{i=1}^{n} \mathbf{x}_i$. *Then the polynomial* $F(\mathbf{x})$ *defined in Eq. (1) satisfies* $F(\mathbf{x}) = sign(\mathbf{x})$ *if* $p > n$.

*Proof.* By Fermat's Little Theorem, for any prime $p > n$, the indicator term $1 - (\mathbf{x} - \mathbf{m})^{p-1} \pmod{p}$ evaluates to 1 if $\mathbf{x} = \mathbf{m}$, and 0 otherwise. Hence, in the summation of Eq. (1), all terms vanish except the one satisfying $\mathbf{x} = \mathbf{m}$. Therefore, we obtain $F(\mathbf{x}) = \text{sign}\left(\sum_{i=1}^{n} \mathbf{x}_i\right) = \text{sign}(\mathbf{x})$, which coincides with the standard majority vote result. $\square$

Once the number of users $n$ and the tie-breaking policy are specified, the majority vote polynomial $F(\mathbf{x})$ can be systematically constructed and efficiently precomputed according to Eq. (1). Representative precomputed polynomials are listed in Table 4 in Appendix D.

### 2.2.2 SECURE EVALUATION OF MAJORITY VOTE POLYNOMIAL $F(\mathbf{x})$

In the FL setting, each user holds a private input $\mathbf{x}_i$ (e.g., sign gradient), and the goal is to securely evaluate a majority vote polynomial $F(\mathbf{x})$ over the aggregated value $\mathbf{x} = \sum_{i=1}^{n} \mathbf{x}_i$, without revealing any input $\mathbf{x}_i$. To perform secure multiplications during polynomial evaluation, we employ additive secret sharing techniques, instantiated for example via Beaver triples (Beaver, 1991), as a practical realization.

For simplicity, we omit the $(\text{mod } p)$ operation and the explicit coefficients of $F(\mathbf{x})$. Let $\deg(F)$ denote the degree of $F(\mathbf{x})$ over $\mathbb{F}_p$. In the offline (initialization) phase, the users collaboratively generate Beaver triples $\{([\![\mathbf{a}^r]\!]_i, [\![\mathbf{b}^r]\!]_i, [\![\mathbf{c}^r]\!]_i) : r \in [R]\}$ via MPC, and each user locally retains its own share, where $R$ is the number of multiplications for securely evaluating the majority vote polynomial. During the online phase (*subround*) for secure polynomial evaluation, each user $i$ recursively computes the shares $[\![\mathbf{x}^k]\!]_i$ of powers $\mathbf{x}^k$ for $k = 1, 2, \ldots, \deg(F)$ as

$$[\![\mathbf{x}^k]\!]_i = \begin{cases} \mathbf{x}_i, & k = 1, \\ [\![\mathbf{c}^r]\!]_i + \boldsymbol{\delta}_{k-v_k}^r \cdot [\![\mathbf{b}^r]\!]_i + \boldsymbol{\epsilon}_{v_k}^r \cdot [\![\mathbf{a}^r]\!]_i + \boldsymbol{\delta}_{k-v_k}^r \cdot \boldsymbol{\epsilon}_{v_k}^r, & k > 1, \end{cases} \quad (2)$$

where $v_k = \max\{j \in \mathbb{N} \mid 2^j \leq k - 1\}$ and $(\boldsymbol{\delta}_{k-v_k}^r, \boldsymbol{\epsilon}_{v_k}^r)$ are obtained by aggregating the masked differences. A fresh Beaver triple is consumed for each multiplication, ensuring that higher-order terms of $F(\mathbf{x})$ are securely computed without exposing any individual input.

The ***subround procedure*** for the secure evaluation of $F(\mathbf{x})$ within the FL framework is as follows:

**Step 1) Evaluation of Shares $[\![\mathbf{x}^k]\!]_i$ for $k = 2$ to $\deg(F)$:**

- For each $k$, each user $i$ computes the masked differences $[\![\mathbf{x}^{k-v_k}]\!]_i - [\![\mathbf{a}^r]\!]_i$ and $[\![\mathbf{x}^{v_k}]\!]_i - [\![\mathbf{b}^r]\!]_i$ based on Eq. (2), and sends them to Server.
- Server aggregates the received masked values to compute: $\boldsymbol{\delta}_{k-v_k}^r = \sum_{i=1}^{n}([\![\mathbf{x}^{k-v_k}]\!]_i - [\![\mathbf{a}^r]\!]_i) = \mathbf{x}^{k-v_k} - \mathbf{a}^r$ and $\boldsymbol{\epsilon}_{v_k}^r = \sum_{i=1}^{n}([\![\mathbf{x}^{v_k}]\!]_i - [\![\mathbf{b}^r]\!]_i) = \mathbf{x}^{v_k} - \mathbf{b}^r$, and broadcasts both $\boldsymbol{\delta}_{k-v_k}^r$ and $\boldsymbol{\epsilon}_{v_k}^r$ to all users.

**Step 2) Local Polynomial Encryption:** Using all received pairs $\{(\boldsymbol{\delta}_{k-v_k}^r, \boldsymbol{\epsilon}_{v_k}^r) : k = 2, \ldots, \deg(F), r \in [R]\}$, each user $i$ locally computes its encrypted share of the evaluated $F(\mathbf{x})$ as:

$$Enc(\mathbf{x}_i) = [\![F(\mathbf{x})]\!]_i = \sum_{k=2}^{\deg(F)} \sum_{r=1}^{R} \left([\![\mathbf{c}^r]\!]_i + \boldsymbol{\delta}_{k-v_k}^r \cdot [\![\mathbf{b}^r]\!]_i + \boldsymbol{\epsilon}_{v_k}^r \cdot [\![\mathbf{a}^r]\!]_i + \boldsymbol{\delta}_{k-v_k}^r \cdot \boldsymbol{\epsilon}_{v_k}^r\right) + \mathbf{x}_i \pmod{p}. \quad (3)$$

The overall encryption procedure is summarized in Algorithm 1, which covers only the user-side encryption steps based on Beaver triples according to the subround in FL framework. For a concrete illustration, see Appendix C.

### 2.3 SECURE MULTIPLICATION-BASED FL FRAMEWORK

In this section, we introduce a novel FL framework that integrates secure aggregation via secure multiplications to preserve user privacy while maintaining aggregation correctness. To clarify the internal mechanisms of the proposed framework, we first describe the key update procedures executed by the users and the central server, respectively.

---

**Algorithm 1** Encryption of Majority Vote Polynomial $F(\mathbf{x})$ via Secure Multiplication (*Subround*)

---

1: **Input:** # selected users $n$, majority vote polynomial $F(\mathbf{x})$, # multiplications $R$
2: **Online Phase**: Encryption of majority vote polynomial $F(\mathbf{x})$
3:   **for** $k = 2$ to $\deg(F)$ **do**                                  (subrounds for evaluating the shares)
4:     **[On User $i$] compute** $[\![\mathbf{x}^{k-v}]\!]_i - [\![\mathbf{a}^r]\!]_i$ and $[\![\mathbf{x}^v]\!]_i - [\![\mathbf{b}^r]\!]_i$, and **send** them **to** Server.
5:     **[On Server] aggregate** masked values to obtain $\delta_k^r$, $\epsilon_k^r$, and **broadcast** them **to** all users.
6:   **end for**
7:   **[On User $i$] compute** secret share $[\![F(\mathbf{x})]\!]_i$ using Eq. (3).
8: **Output:** (Generation of $[\![F(\mathbf{x})]\!]_i$ for user $i$) $Enc(\mathbf{x}_i) = [\![F(\mathbf{x})]\!]_i$

---

**Algorithm 2** Secure Majority Vote Aggregation via Secure Multiplication

---

1: **Input:** Initial model $\boldsymbol{\theta}_0$, learning rate $\eta$, # selected users $n$, majority vote polynomial $F(\mathbf{x})$
2: **for** $t = 0$ to $T - 1$ **do**
3:   **[On User $i$]**
4:     **compute** local gradient: $\mathbf{g}_i(t)$
5:     **quantize** gradient: $\mathbf{x}_i(t) = q(\mathbf{g}_i(t)) \in \{-1, 1\}^d$
6:     **generate** secret share: $Enc(\mathbf{x}_i(t)) \leftarrow [\![F(\mathbf{x}(t))]\!]_i$ using Algorithm 1 for $\mathbf{x}(t) = \sum_{i=1}^n \mathbf{x}_i(t)$
7:     **transmit** $Enc(\mathbf{x}_i(t))$ **to** Server
8:   **[On Server]**
9:     **aggregate** encrypted shares: $F(\mathbf{x}(t)) = \sum_{i=1}^n Enc(\mathbf{x}_i(t))$              (see Eq. (5))
10:     **obtain** majority vote: $\tilde{\mathbf{g}}(t) = \text{sign}\left(\sum_{i=1}^n \mathbf{x}_i(t)\right) \leftarrow F(\mathbf{x}(t))$
11:     **broadcast** $\tilde{\mathbf{g}}(t)$ **to** all users
12:   **[On User $i$] update** model: $\boldsymbol{\theta}(t + 1) \leftarrow \boldsymbol{\theta}(t) - \eta\tilde{\mathbf{g}}(t)$
13: **end for**
14: **Output:** $\boldsymbol{\theta}(T)$

---

### 2.3.1 USER UPDATE PROCEDURE

**Step 1 (Sign Gradient Calculation):** The user $i$ computes the gradient $\mathbf{g}_i(t)$ using the global model $\boldsymbol{\theta}(t)$ and performs 1-bit quantization to obtain the locally updated sign gradient $\mathbf{x}_i(t)$:

$$\mathbf{x}_i(t) = \text{sign}(\mathbf{g}_i(t)), \quad \mathbf{x}_i(t) \in \{-1, 1\}^d \tag{4}$$

where $d$ denotes the size of the global model.

**Step 2 (Secure Evaluation of $[\![F(\mathbf{x})]\!]_i$):** At subround, each user $i$ employs Beaver triples, as an example instantiation of secure multiplication, pre-distributed to securely evaluate its share of the majority vote polynomial $F(\mathbf{x}(t))$ for $\mathbf{x}(t) = \sum_{i=1}^n \mathbf{x}_i(t)$, represented as $[\![F(\mathbf{x}(t))]\!]_i$. This share is then used to compute the sign of the aggregated input vectors. Notably, this computation is performed without revealing any individual input $\mathbf{x}_i(t)$. Further details are provided in Section 2.2 and Algorithm 1. Finally, each user $i$ sends its encrypted share of the majority vote polynomial to the server: $Enc(\mathbf{x}_i(t)) = [\![F(\mathbf{x}(t))]\!]_i$.

### 2.3.2 MODEL AGGREGATION PROCEDURE

From the encrypted local updates $\{Enc(\mathbf{x}_i(t)) : i \in [n]\}$, Server computes the final majority vote result $\tilde{\mathbf{g}}(t)$ and broadcasts it to all users as follows:

**Aggregation:** $$F(\mathbf{x}(t)) = \sum_{i=1}^n Enc(\mathbf{x}_i(t)) = \sum_{i=1}^n [\![F(\mathbf{x}(t))]\!]_i \pmod{p}, \tag{5}$$

where $\mathbf{x}(t) = \sum_{i=1}^n \mathbf{x}_i(t)$ and $\sum_{i=1}^n [\![F(\mathbf{x}(t))]\!]_i = \text{sign}(\mathbf{x}(t))$.

**Broadcasting:** $$\tilde{\mathbf{g}}(t) = F(\mathbf{x}(t)) = \text{sign}(\mathbf{x}(t)). \tag{6}$$

The overall aggregation procedure is summarized in Algorithm 2. Each user encrypts and transmits its share of the majority vote polynomial $F(\mathbf{x})$, while the server aggregates the received shares as in Eq. (5) and broadcasts the resulting global direction $\tilde{\mathbf{g}}(t)$ to all users for model update.

## 2.4 SUBGROUP-BASED SECURE FL FRAMEWORK

To mitigate the overhead of securely evaluating the majority vote polynomial $F(\mathbf{x})$ with secure multiplication techniques (e.g., Beaver triples, DN, ATLAS), whose cost grows significantly with the number of users, we propose a subgrouping strategy that partitions users into smaller subsets $\mathcal{G}_j$. Each subgroup securely aggregates its inputs independently, and the final result is obtained by combining all subgroup outputs. This reduces the polynomial degree, latency, and bandwidth, while ensuring scalability and privacy with manageable computational and communication costs.

**Subgrouping and Hierarchical Majority Vote Aggregation:** As the number of users $n$ increases, the degree of the majority vote polynomial $F(\mathbf{x})$ also grows, which raises the number of secure multiplication subrounds required for polynomial evaluation. This results in higher uplink communication cost and latency, thereby limiting scalability. In addition, a larger prime modulus $p > n$ must be chosen, which further increases the complexity of evaluating $F(\mathbf{x})$ over $\mathbb{F}_p$. To address these limitations, we propose a subgrouping strategy that partitions the total $n$ users into $\ell$ disjoint subgroups, each of size $n_1 = n/\ell$. Within each subgroup, a small majority vote polynomial is evaluated independently based on local inputs. Since the polynomial degree now depends on the smaller subgroup size $n_1$, the number of required secure subrounds is reduced and a smaller prime modulus $p_1(> n_1)$ is adopted. Consequently, subgrouping leads to significant reductions in both computational and communication costs. Nevertheless, additional protection of subgroup outputs is required to ensure end-to-end privacy. We further analyze the impact of tie-breaking policies and hierarchical aggregation on communication and computation, as detailed in Appendix G.

The proposed aggregation procedure is executed in two hierarchical steps:

**Step 1 (Intra-subgroup Majority Vote):** Within each subgroup $\mathcal{G}_j, j \in [\ell]$, the local majority vote is securely evaluated as

$$F(\mathbf{x}_j(t)) = \sum_{i=1}^{n_1} Enc(\mathbf{x}_{i,j}(t)) = \sum_{i=1}^{n_1} [\![F(\mathbf{x}_j(t))]\!]_i \pmod{p_1}, \tag{7}$$

where $\sum_{i=1}^{n_1} [\![F(\mathbf{x}_j(t))]\!]_i = \mathrm{sign}\,(\mathbf{x}_j(t))$ for $\mathbf{x}_j(t) = \sum_{i=1}^{n_1} \mathbf{x}_{i,j}(t)$.

**Step 2 (Inter-subgroup Majority Vote):** The global majority vote is computed by aggregating the results across all subgroups:

$$\tilde{\mathbf{g}}(t) = \mathrm{sign}\bigg( \sum_{j=1}^{\ell} F\big(\mathbf{x}_j(t)\big) \bigg) = \mathrm{sign}\bigg( \sum_{j=1}^{\ell} \mathrm{sign}\bigg( \sum_{i=1}^{n_1} \mathbf{x}_{i,j}(t) \bigg) \bigg). \tag{8}$$

The global majority vote result $\tilde{\mathbf{g}}(t)$ is subsequently broadcast to all users. The overall protocol is summarized in Algorithm 3.

---

**Algorithm 3** Hierarchical Secure Majority Vote Aggregation with Subgrouping

---

1: **Input:** Initial model $\boldsymbol{\theta}_0$, learning rate $\eta$, # selected users $n$, # subgroups $\ell$, majority vote polynomial $F(\mathbf{x}_j)$ for each subgroup
2: **for** $t = 0$ to $T - 1$ **do**
3:     **[On User $i$ in subgroup $\mathcal{G}_j$]**
4:       **compute** local gradient: $\mathbf{g}_{i,j}(t)$
5:       **quantize** gradient: $\mathbf{x}_{i,j}(t) = q(\mathbf{g}_{i,j}(t)) \in \{-1, 1\}^d$
6:       **generate** secret share: $Enc(\mathbf{x}_{i,j}(t)) \leftarrow [\![F(\mathbf{x}_j(t))]\!]_i$ for $\mathbf{x}_j(t) = \sum_{i=1}^{n_1} \mathbf{x}_{i,j}(t)$ using Algorithm 1
7:       **transmit** $Enc(\mathbf{x}_{i,j}(t))$ **to** Server
8:     **[On Server]**
9:       **reconstruct** $F(\mathbf{x}_j(t))$ **from** received shares for each subgroup $\mathcal{G}_j$          (see Eq. (7))
10:       **compute** global vote: $\tilde{\mathbf{g}}(t) = \mathrm{sign}(\sum_{j=1}^{\ell} F(\mathbf{x}_j(t))) \in \{-1, 1\}^d$
11:       **broadcast** $\tilde{\mathbf{g}}(t)$ **to** all users
12:     **[On User $i$] update** model: $\boldsymbol{\theta}(t + 1) \leftarrow \boldsymbol{\theta}(t) - \eta\tilde{\mathbf{g}}(t)$
13: **end for**
14: **Output:** $\boldsymbol{\theta}(T)$

---

# 3 THEORETICAL ANALYSIS

In this section, we present an analysis of the convergence and the security properties of the proposed Hi-SAFE method. The main theoretical results are proved in Appendices E and F, which also provide a formal proof of corruption tolerance along with additional proofs that support the main analysis. Furthermore, we provide a computational complexity analysis and runtime evaluation of the proposed method in Appendix H.

## 3.1 CONVERGENCE ANALYSIS

We analyze the convergence of the proposed Hi-SAFE, a SIGNSGD algorithm with hierarchical majority vote. The analysis builds upon standard stochastic optimization assumptions and extends the convergence result of (Bernstein et al., 2018a) to a hierarchical subgrouping framework.

Let $n$ users be partitioned into $\ell$ subgroups of equal size $n_1 = n/\ell$. Suppose that each subgroup outputs the correct majority vote per coordinate with probability $q \in (0.5, 1]$, independently across subgroups. Let $f^\star$ and $f_0$ be the minimum and initial values of the global objective, respectively, and let $\vec{L}$ and $\vec{\sigma}$ denote the coordinate-wise smoothness and variance bound vectors, respectively.

**Theorem 1** (Convergence of SIGNSGD with Hierarchical Majority Vote). *Run Algorithm 3 for $K$ iterations with learning rate $\eta = 1/\sqrt{K\|\vec{L}\|_1}$, mini-batch size $m_k = K$, and let $N_t$ denote the total number of stochastic gradient evaluations per user. Then the algorithm achieves the following bound:*

$$\mathbb{E}\left[\frac{1}{K}\sum_{k=0}^{K-1}\|\mathbf{g}_k\|_1\right]^2 \leq \frac{1}{\sqrt{N_t}}\left(\sqrt{\|\vec{L}\|_1}(f_0 - f^\star + \tfrac{1}{2}) + \frac{2}{\sqrt{n_1}}\|\vec{\sigma}\|_1 + c'\ell^{-1/4}\exp\left(-\frac{\ell\alpha_q^2}{4}\right)\right)^2,$$

*where $\alpha_q := \frac{(2q-1)}{2\sqrt{q(1-q)}} > 0$ and $c' > 0$ is a constant depending on global aggregation perturbations.*

**Remark 1** (Convergence–Communication Trade-off). *Subgrouping offers a flexible trade-off: fewer subgroups (larger $n_1$) yield lower variance and faster convergence, while more subgroups reduce per-user communication and support scalable deployment.*

**Remark 2** (Subgrouping under $q > 1/2$). *If each subgroup outputs the correct majority vote with probability $q > 1/2$, then the global aggregation error decays as $\exp(-2\ell(q-0.5)^2)$, local robustness improves with accuracy gap scaling as $\mathcal{O}(1/\sqrt{n_1})$, and the hierarchical structure reduces per-user communication compared to the non-subgrouping scheme. Subgrouping thus enables an effective trade-off between convergence and communication efficiency.*

## 3.2 SECURITY ANALYSIS

We show that Hi-SAFE preserves input privacy under the semi-honest model with deterministic tie-breaking $\text{sign}(0) = \tau \in \{-1, 0, +1\}$. The server learns nothing beyond the final majority $\mathbf{s}$.

Consider $n$ users partitioned into $\ell$ subgroups $\{\mathcal{G}_j\}_{j=1}^{\ell}$ of size $n_1 = n/\ell$. Each user $i \in \mathcal{G}_j$ holds $\mathbf{x}_{i,j} \in \{-1, +1\}^d$, with subgroup aggregate $\mathbf{x}_j = \sum_{i=1}^{n_1}\mathbf{x}_{i,j}$ and majority $\mathbf{s}_j = \text{sign}(\mathbf{x}_j)$. Secure multiplications use Beaver triples with offline randomness independent of inputs. The adversary may corrupt at most $t \leq n-1$ users, with at least one honest user per subgroup. We employ moduli $p_1 > n_1$ for subgroup computation and $p_2 > \ell$ for inter-group aggregation. In the local stage, subgroups output only secret shares $\{[\![F(\mathbf{x}_j)]\!]_i\}$ of the majority polynomial. The global stage securely computes $\mathbf{s} = \text{sign}\left(\sum_{j=1}^{\ell}\mathbf{s}_j\right) = \sum_{j=1}^{\ell}\sum_{i=1}^{n_1}[\![F(\mathbf{x}_j)]\!]_i$ over $\mathbb{F}_{p_2}$. We denote by $\text{REAL}_{\Pi,\mathcal{A}}$ the adversary's view during protocol execution, including corrupted inputs, randomness, and messages. $\text{SIM}_{\mathcal{A}}$ denotes the output of a PPT simulator with access only to corrupted inputs and the final result $\mathbf{s}$.

**Theorem 2** (Security of Hi-SAFE Aggregation). *For any PPT semi-honest adversary $\mathcal{A}$ corrupting at most $t \leq n-1$ users, let $\mathcal{C} \subseteq [n]$ be the set of corrupted users. Then there exists a PPT simulator $\text{SIM}$ such that*

$$\text{REAL}_{\Pi,\mathcal{A}}(\{\mathbf{x}_{i,j}\}_{i \in \mathcal{C}}) \approx_c \text{SIM}_{\mathcal{A}}(\{\mathbf{x}_{i,j}\}_{i \in \mathcal{C}}, \mathbf{s}),$$

*where $\approx_c$ denotes the computational indistinguishability.*

**Remark 3** (Residual Leakage Probability). *If each $\mathbf{x}_{i,j}$ is chosen independently and uniformly at random, the only case where inputs can be inferred from the final majority vote result $\mathbf{s}$ is when all $n$ inputs are identical, which occurs with probability $\Pr[\text{all inputs identical}] = 2 \cdot \left(\frac{1}{2}\right)^n = \frac{1}{2^{n-1}}$. Thus, input privacy failure occurs with negligible probability $O(2^{-n})$. Unlike masking-based methods, which reveal intermediate sums and fully expose inputs in such extreme cases, Hi-SAFE keeps all intermediate computations secret-shared and reveals only the final majority vote result $\mathbf{s}$, ensuring no additional leakage even under extreme input distributions.*

## 4 EXPERIMENTS

### 4.1 EXPERIMENT SETUP

To evaluate the effectiveness and practicality of the proposed HiSAFE, we conducted experiments on multiple benchmark datasets, including MNIST (LeCun et al., 1998), FMNIST (Xiao et al., 2017), and CIFAR-10 (Krizhevsky et al., 2009). Our experiments are conducted on a GPU server with 2 NVIDIA RTX 3090 GPUs. Each experiment is executed three independent trials with distinct random seeds to calculate average metrics and ensure the reproducibility of our results. The detailed training parameters are described in Appendix I.1.

### 4.2 EXPERIMENT RESULTS

Figure 2 compares the model performance under different tie-breaking policies for non-subgrouping and optimal subgrouping with $n = 24$ users. Figure 2a shows the baseline setting in which 1-bit tie-breaking is applied to both intra- and inter-subgroup aggregation, while Figure 2b applies 2-bit tie-breaking only to intra-subgroup aggregation. The experimental results indicate that both 1-bit and 2-bit tie-breaking strategies yield comparable model accuracy, each exhibiting distinct trade-offs. While 1-bit tie-breaking reduces computational complexity, 2-bit tie-breaking[1] slightly reduces the number of terms in $F(\mathbf{x})$ and improves computational precision, albeit at the cost of increased server-side complexity. Accordingly, the choice between the two strategies should be guided by the desired balance between computational efficiency and model accuracy. Hi-SAFE achieves comparable performance under 1-bit tie-breaking and improved accuracy under 2-bit tie-breaking, owing to enhanced computational precision on the server side—especially when subgrouping is applied. Note that, under the 1-bit tie-breaking setting, the non-subgrouping configuration of Hi-SAFE is equivalent to naive SIGNSGD-MV, except for its privacy guarantees. To support these findings, additional experimental results under various FL settings are provided in Appendix I.2.

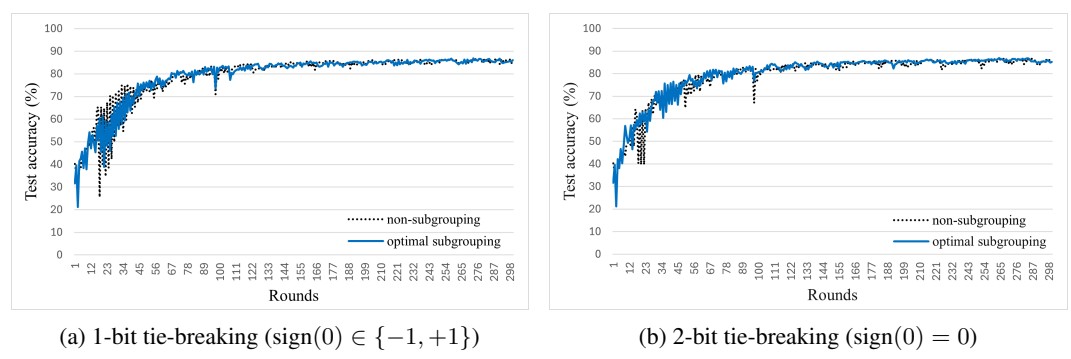

(a) 1-bit tie-breaking ($\text{sign}(0) \in \{-1, +1\}$)         (b) 2-bit tie-breaking ($\text{sign}(0) = 0$)

Figure 2: Performance comparison of different tie-breaking policies on the FMNIST dataset.

### 4.3 EVALUATION OF OPTIMAL SUBGROUPING STRATEGY

Table 1 summarizes the optimal subgroup configurations $\ell^\star$ that minimize the total communication cost $C_T$ for various user counts $n$. Parentheses denote the percentage reduction relative to the

---

[1] Since intra-subgroup computations are performed entirely on the server side, they incur no additional uplink communication cost (refer to Appendix G).

Table 1: Optimal subgroup configuration and communication cost

| $n$ | $\ell^{\star}$ | $n_1$ | $\lceil \log p_1 - 1 \rceil$ | #multiplications | $C_T$ (%) | $C_u$ (%) |
|---|---|---|---|---|---|---|
| 24 | 8 | 3 | 2 | 4 | 96(52.0%) | 12(94.0%) |
| 36 | 12 | 3 | 2 | 4 | 144(47.8%) | 12(95.7%) |
| 60 | 20 | 3 | 2 | 4 | 240(44.4%) | 12(97.2%) |
| 90 | 30 | 3 | 2 | 4 | 360(50.5%) | 12(98.4%) |
| 100 | 25 | 4 | 2 | 6 | 450(43.6%) | 18(97.7%) |

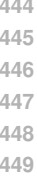
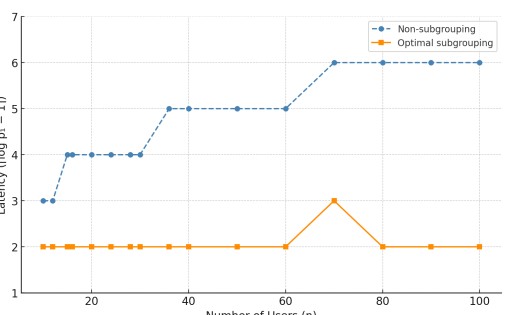

(a) Per-user secure multiplications      (b) Latency $\lceil \log p_1 - 1 \rceil$ for secure multiplication

Figure 3: Impact of optimal subgrouping on secure multiplication cost and latency.

non-subgrouping baseline. These results demonstrate that Hi-SAFE achieves substantial reductions in both total and per-user communication costs ($C_u$) without degrading model accuracy. Notably, for $n \geq 24$, the per-user communication cost consistently decreases by more than 94%, with up to 52.0% reduction in total communication cost observed at $n = 24$. These findings validate the scalability and communication efficiency of the proposed framework.

Figure 3 illustrates the effect of optimal subgrouping on per-user secure multiplications and their latency. In the non-subgrouping setting (Figure 3a), a global majority vote polynomial must be evaluated over all $n$ users, resulting in a per-user cost that increases linearly with $n$. In contrast, the proposed subgrouping strategy partitions users into subgroups of size $n_1$, enabling each group to evaluate a small majority vote polynomial. This approach keeps the per-user cost constant and low ($\leq 6$), regardless of system scale. Figure 3b shows the latency, defined as the serial depth $\lceil \log p_1 - 1 \rceil$ for Beaver triple multiplication. In the non-subgrouping case, larger finite fields are needed to support global majority vote polynomials, which results in increased latency. Subgrouping, on the other hand, confines computation to smaller groups, enabling the use of smaller fields and consistently achieving low latency—often as low as 2. To complement these findings, we further evaluate the effect of varying subgrouping configurations, parameterized by $\ell$, in Appendix I.3.

## 5 CONCLUSION

In this paper, we have proposed Hi-SAFE, a lightweight and cryptographically secure aggregation framework for communication-efficient and privacy-preserving FL. By securely evaluating majority vote polynomials under additive secret sharing, instantiated for example via Beaver triples, Hi-SAFE achieves end-to-end privacy for sign-based FL, revealing only the final majority vote to the server under the semi-honest model. Furthermore, the proposed hierarchical subgrouping strategy ensures constant latency and a bounded secure multiplication cost per user, independent of the total number of users. Extensive theoretical and experimental analyses demonstrate that Hi-SAFE reduces per-user communication cost by over 94% when $n \geq 24$, and achieves up to 52% reduction in total communication cost at $n = 24$, while preserving model accuracy. These results confirm the scalability, robustness, and practicality of Hi-SAFE, especially in bandwidth-constrained FL deployments.

## REPRODUCIBILITY STATEMENT

To ensure reproducibility, we include the complete source code in the supplementary material. The main text describes the overall methodology and experimental setup, while the appendix provides detailed proofs of theoretical claims and extended evaluations. Furthermore, the appendix contains additional experimental results across diverse environments and datasets, as well as a full description of dataset preprocessing steps and instructions for reproducing the reported results. We also provide explanatory documentation within the codebase to facilitate understanding and reuse by other researchers in the supplementary material.

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

## LLM USAGE STATEMENT

In accordance with the ICLR policy on the use of large language models (LLMs), we describe here the role of LLMs in the preparation of this work. LLMs were used only for grammar checking and expression refinement in certain paragraphs. They were not involved in research ideation, algorithm design, theoretical analysis, or experimental implementation. All core contributions, including the problem formulation, method development, mathematical proofs, experimental setup, and result analysis, are original work of the authors.

## A   NOTATION

The notations used throughout this paper are summarized in Table 2. Standard mathematical symbols are employed unless otherwise specified.

Table 2: Summary of frequently used notations

| Notation | Description |
|---|---|
| $\mathbb{F}_p$ | Finite field of prime order $p$ |
| $\mod p$ | Modulo operation over a prime $p$ |
| $[n]$ | Index set $\{1, 2, \ldots, n\}$ |
| $x, y$ | Scalars (denoted in regular lowercase letters) |
| $\mathbf{x}, \mathbf{y}$ | Vectors (denoted in bold lowercase letters) |
| $f(\boldsymbol{\theta})$ | Global objective function evaluated at $\boldsymbol{\theta}$ |
| $f^{\star}$ | Minimum value of the global objective function |
| $\vec{L}$ | Smoothness vector $[L_1, \ldots, L_d]$ for function $f$ |
| $\vec{\sigma}$ | Variance bound vector for stochastic gradients |
| $[\![x]\!]_i$ | Share of $x$ for user $i$ |
| $\|\cdot\|_1$ | $\ell_1$-norm |
| $C_u$ | Per-user communication cost |
| $C_T$ | Total communication cost |
| $R$ | Total number of secure multiplications |
| $\mathbb{E}[\cdot]$ | Expectation operator for random variables |

## B   RELATED WORK

Numerous secure aggregation strategies have been developed to mitigate privacy risks in FL, including masking, DP, and HE. While each of these methods provides certain privacy guarantees, they exhibit significant limitations concerning communication efficiency, compatibility with sign-based protocols, and robustness against inference attacks.

Masking-based methods (Bonawitz et al., 2017; So et al., 2022) typically employ pairwise secret sharing to cryptographically protect individual updates while ensuring correct aggregation. Although these methods are scalable, they expose intermediate aggregation results to the server or auxiliary nodes, potentially leading to information leakage under semi-honest assumptions unless additional mechanisms, such as double masking, are employed. Local DP methods (Truex et al., 2019; Byrd & Polychroniadou, 2020; Lyu, 2021) perturb local model updates with noise prior to aggregation, thereby providing formal privacy guarantees. For instance, DP-SIGNSGD (Lyu, 2021) adds Gaussian noise before applying the sign function. However, the presence of noisy sign gradients remains visible to the server, and achieving strong privacy often requires substantial noise, which can degrade model accuracy—especially problematic in data-sparse IoT environments. HE (Cheon et al., 2017; Zhang et al., 2020; Fang & Qian, 2021; Jiang et al., 2021; Ma et al., 2022; Gentry, 2009) allows for computation directly over encrypted data, offering strong cryptographic security. However, HE-based schemes incur significant computational overhead and produce large ciphertexts (e.g., thousands of bits per coordinate), which render them impractical for bandwidth-limited FL deployments. Additionally, HE does not support nonlinear functions, such as the sign function or majority vote, which are essential for the functionality of sign-based protocols like SIGNSGD-MV.

Despite their strengths, existing secure aggregation methods are not directly compatible with sign-based protocols. Specifically, masking-based approaches permit the server to access intermediate summation values during the computation of the final majority vote, which may result in information leakage. HE-based schemes are fundamentally incompatible with nonlinear vote operations. Moreover, the high communication cost associated with HE undermines the primary advantage of SIGNSGD-MV—its 1-bit update efficiency. **A detailed comparison is provided in Appendix B.1**.

To address these limitations, we propose Hi-SAFE, a cryptographic secure aggregation framework for SIGNSGD-MV. Hi-SAFE privately evaluates majority votes via secure multiplications, thereby preserving communication efficiency and enabling scalable, privacy-preserving FL under the semi-honest model.

## B.1 Comparison with Existing Secure Aggregation Methods

Table 3 provides a comparative summary of the proposed Hi-SAFE framework and existing secure aggregation methods in FL. The comparison considers multiple criteria including the type of privacy guarantee, exposure level to the server, accuracy preservation, and overall communication and computational efficiency.

As summarized, existing methods such as masking and local DP offer partial protection but exhibit key limitations when applied to sign-based protocols. In particular, masking-based approaches expose intermediate summation values during majority vote computation, and DP schemes suffer from accuracy degradation due to the addition of noise. HE, while cryptographically strong, is computationally intensive and fundamentally incompatible with nonlinear operations such as $\text{sign}(\cdot)$ or majority vote.

By contrast, the proposed Hi-SAFE framework achieves privacy-preserving aggregation tailored to 1-bit SIGNSGD-MV by securely evaluating majority vote polynomials, instantiated for example via Beaver triples. It reveals only the final majority vote result, preserves communication efficiency, and scales well under semi-honest assumptions. Moreover, unlike masking-based methods that fully leak inputs in extreme cases (e.g., all users submit $-1$ or all submit $+1$), Hi-SAFE prevents such leakage by keeping all intermediate computations secret-shared. Under a uniform input distribution, the probability of accidental input privacy loss is at most $1/2^{n-1}$, which is negligible in the number of users $n$.

Table 3: Comparison of the proposed method with the existing privacy-preserving aggregation approaches

| Method | Privacy Type | Server Observes | Accuracy Loss | Comm. Efficiency | Comp. Cost | Scalability |
|---|---|---|---|---|---|---|
| Masking (Bonawitz et al., 2017) | Cryptographic (Double Masking) | ✓ (Summation Values) | ✗ | Low | High | Limited |
| DP (Lyu, 2021) | Formal (Local DP) | ✓ (Noisy Sign Gradients) | ✓ (High) | High | Low | High |
| HE (Cheon et al., 2017) | Cryptographic (RLWE-based HE) | ✗ (Fully Encrypted) | ✗ | Very Low | Very High | Very Limited |
| SIGNSGD-MV (Bernstein et al., 2018a) | - | ✓ (All Raw Sign Gradients) | ✗ | Very High | Very Low | Very High |
| **Proposed Method** | **Cryptographic (Beaver triples)** | **✓ (Final Majority Vote Only)** | ✗ | **High** | **Low** | **High** |

## C Illustrative Example: Secure Evaluation of the Majority Vote Polynomial

To aid understanding, we present a step-by-step example of securely computing the majority vote polynomial $F(\mathbf{x})$ using Beaver triples. In particular, we describe the detailed procedure by which each user $i$ securely evaluates an encrypted share $[\![F(\mathbf{x})]\!]_i$, which is subsequently transmitted to Server for aggregation.

## C.1 SECURE EVALUATION OF $F(\mathbf{x}) = 2\mathbf{x}^3 + 4\mathbf{x} \pmod 5$ WITH $n = 3$

As a concrete example, consider the evaluation of the polynomial $F(\mathbf{x}) = 2\mathbf{x}^3 + 4\mathbf{x} \pmod 5$ over the finite field $\mathbb{F}_5$, assuming $n = 3$ users. Each user holds a private input $\mathbf{x}_i \in \{-1, +1\}$ such that:

$$\mathbf{x} = \sum_{i=1}^{3} \mathbf{x}_i.$$

For simplicity, we assume the following scalar user inputs:

$$x_1 = 1, \quad x_2 = -1, \quad x_3 = 1,$$

which yield the majority vote result:

$$\mathrm{sign}\left(\sum_{i=1}^{3} x_i\right) = \mathrm{sign}(1 - 1 + 1) = \mathrm{sign}(1) = 1.$$

To evaluate $F(x)$ securely, we adopt a Beaver triple-based protocol. In this example, the following Beaver triples are pre-shared among users during the offline phase:

$$a^r = \sum_{i=1}^{3} [\![a^r]\!]_i, \quad b^r = \sum_{i=1}^{3} [\![b^r]\!]_i, \quad c^r = a^r \cdot b^r = \sum_{i=1}^{3} [\![c^r]\!]_i, \quad r \in \{1, 2\},$$

with each share lying in the field $\mathbb{F}_5$. The specific shares are given by:

$$[\![a^1]\!]_1 = 0, \quad [\![a^1]\!]_2 = 3, \quad [\![a^1]\!]_3 = 2, \quad [\![a^2]\!]_1 = 4, \quad [\![a^2]\!]_2 = 3, \quad [\![a^2]\!]_3 = 1,$$
$$[\![b^1]\!]_1 = 2, \quad [\![b^1]\!]_2 = 2, \quad [\![b^1]\!]_3 = 0, \quad [\![b^2]\!]_1 = 0, \quad [\![b^2]\!]_2 = 1, \quad [\![b^2]\!]_3 = 4,$$
$$[\![c^1]\!]_1 = 1, \quad [\![c^1]\!]_2 = 1, \quad [\![c^1]\!]_3 = 3, \quad [\![c^2]\!]_1 = 1, \quad [\![c^2]\!]_2 = 2, \quad [\![c^2]\!]_3 = 2.$$

To evaluate $F(x)$ securely, it is necessary to first compute the shared cubic term $x^3$, which can be decomposed as:

$$\begin{aligned}
x^3 = x \cdot x^2 &= (x - a^2 + a^2)(x^2 - b^2 + b^2) \\
&= (x - a^2)(x^2 - b^2) + a^2(x^2 - b^2) + b^2(x - a^2) + c^2 \pmod 5,
\end{aligned} \tag{9}$$

where $x^2$ itself is computed via:

$$\begin{aligned}
x^2 = x \cdot x &= (x - a^1 + a^1)(x - b^1 + b^1) \\
&= (x - a^1)(x - b^1) + a^1(x - b^1) + b^1(x - a^1) + c^1 \pmod 5.
\end{aligned} \tag{10}$$

As the computation of $x^3$ requires the intermediate value $x^2$, the evaluation proceeds in two secure multiplication subrounds. We now describe each subround in detail.

- **In subround 0:** each user $i$ prepares the necessary values for computing $x^2$ using Beaver triples. Specifically, each user locally computes the following masked values:

$$(x_i - [\![a^1]\!]_i) \quad \text{and} \quad (x_i - [\![b^1]\!]_i) \pmod 5.$$

The local computations for each user are given below:

$$\begin{aligned}
\text{User 1:} \quad & x_1 - [\![a^1]\!]_1 = 1 - 0 = 1 \pmod 5, \\
& x_1 - [\![b^1]\!]_1 = 1 - 2 = -1 \equiv 4 \pmod 5, \\
\text{User 2:} \quad & x_2 - [\![a^1]\!]_2 = -1 - 3 = -4 \equiv 1 \pmod 5, \\
& x_2 - [\![b^1]\!]_2 = -1 - 2 = -3 \equiv 2 \pmod 5, \\
\text{User 3:} \quad & x_3 - [\![a^1]\!]_3 = 1 - 2 = -1 \equiv 4 \pmod 5, \\
& x_3 - [\![b^1]\!]_3 = 1 - 0 = 1 \pmod 5.
\end{aligned}$$

Each user transmits the computed masked values to Server, which then aggregates the results to obtain:

$$x - a^1 = \sum_{i=1}^{3}(x_i - [\![a^1]\!]_i) = 1 + 1 + 4 = 6 \equiv 1 \pmod{5},$$

$$x - b^1 = \sum_{i=1}^{3}(x_i - [\![b^1]\!]_i) = 4 + 2 + 1 = 7 \equiv 2 \pmod{5}.$$

Server then broadcasts the aggregated values $(x - a^1)$ and $(x - b^1)$ to all users to complete the secure multiplication step for computing $x^2$.

- **In subround 1:** each user $i$ prepares the values necessary to compute $x^3$ using Beaver triples. To this end, the following quantities must be obtained:

$$(x_i - [\![a^2]\!]_i) \quad \text{and} \quad ([\![x^2]\!]_i - [\![b^2]\!]_i) \pmod{5}.$$

Here, each user computes $[\![x^2]\!]_i$ based on the pre-shared Beaver triples $([\![a^1]\!]_i, [\![b^1]\!]_i, [\![c^1]\!]_i)$ and the publicly computable term $(x - a^1)(x - b^1)$, as defined in Eq. (10):

$$[\![x^2]\!]_i = (x - a^1)(x - b^1) + [\![a^1]\!]_i(x - b^1) + [\![b^1]\!]_i(x - a^1) + [\![c^1]\!]_i \pmod{5}.$$

Since the product $(x - a^1)(x - b^1)$ is constant across all users, only a single user needs to compute and broadcast it to the server. Without loss of generality, we assume that User 1 performs this computation.

Based on the received values, the users compute $[\![x^2]\!]_i$ as follows:

$$\text{User 1:} \quad [\![x^2]\!]_1 = 1 \cdot 2 + 0 \cdot 2 + 2 \cdot 1 + 1 = 5 \equiv 0 \pmod{5},$$
$$\text{User 2:} \quad [\![x^2]\!]_2 = 3 \cdot 2 + 2 \cdot 1 + 1 = 9 \equiv 4 \pmod{5},$$
$$\text{User 3:} \quad [\![x^2]\!]_3 = 3 \cdot 2 + 2 \cdot 1 + 1 = 9 \equiv 4 \pmod{5}.$$

Next, each user computes the masked values required for secure multiplication of $x \cdot x^2$:

$$\text{User 1:} \quad x_1 - [\![a^2]\!]_1 = 1 - 4 = -3 \equiv 2 \pmod{5},$$
$$[\![x^2]\!]_1 - [\![b^2]\!]_1 = 0 - 0 = 0 \pmod{5},$$
$$\text{User 2:} \quad x_2 - [\![a^2]\!]_2 = -1 - 3 = -4 \equiv 1 \pmod{5},$$
$$[\![x^2]\!]_2 - [\![b^2]\!]_2 = 4 - 1 = 3 \pmod{5},$$
$$\text{User 3:} \quad x_3 - [\![a^2]\!]_3 = 1 - 1 = 0 \pmod{5},$$
$$[\![x^2]\!]_3 - [\![b^2]\!]_3 = 4 - 4 = 0 \pmod{5}.$$

Each user then transmits the above masked values to Server. The server aggregates the results to reconstruct the global masked values:

$$x - a^2 = \sum_{i=1}^{3}(x_i - [\![a^2]\!]_i) = 2 + 1 + 0 = 3 \pmod{5},$$

$$x^2 - b^2 = \sum_{i=1}^{3}([\![x^2]\!]_i - [\![b^2]\!]_i) = 0 + 3 + 0 = 3 \pmod{5}.$$

Server then broadcasts these aggregated values $(x - a^2)$ and $(x^2 - b^2)$ to all users to complete the secure multiplication step for computing $x^3$.

**Global computation:** After completing the two subrounds, each user proceeds to compute a share of the final majority vote polynomial $F(x) = 2x^3 + 4x \pmod{5}$. Using the broadcast values $(x - a^2)$ and $(x^2 - b^2)$ from Server, and their local input $x_i$, each user locally evaluates the masked cubic term $[\![x^3]\!]_i$ as follows:

$$[\![x^3]\!]_i = (x - a^2)(x^2 - b^2) + [\![a^2]\!]_i(x^2 - b^2) + [\![b^2]\!]_i(x - a^2) + [\![c^2]\!]_i \pmod{5}.$$

The individual computations are given below:

$$\text{User 1:} \quad [\![x^3]\!]_1 = 3 \cdot 1 + 4 \cdot 1 + 0 \cdot 3 + 1 = 8 \equiv 3 \pmod 5,$$
$$\text{User 2:} \quad [\![x^3]\!]_2 = 3 \cdot 1 + 1 \cdot 3 + 2 = 8 \equiv 3 \pmod 5,$$
$$\text{User 3:} \quad [\![x^3]\!]_3 = 1 \cdot 1 + 4 \cdot 3 + 2 = 15 \equiv 0 \pmod 5.$$

Each user then substitutes the locally computed values into the final polynomial:

$$[\![F(x)]\!]_i = 2[\![x^3]\!]_i + 4x_i \pmod 5.$$

The share computations are as follows:

$$\text{User 1:} \quad [\![F(x)]\!]_1 = 2 \cdot 3 + 4 \cdot 1 = 10 \equiv 0 \pmod 5,$$
$$\text{User 2:} \quad [\![F(x)]\!]_2 = 2 \cdot 3 + 4 \cdot (-1) = 2 \pmod 5,$$
$$\text{User 3:} \quad [\![F(x)]\!]_3 = 2 \cdot 0 + 4 \cdot 1 = 4 \pmod 5.$$

Each user sends their computed share $[\![F(x)]\!]_i$ to Server. The server aggregates the values to obtain the final result:

$$F(x) = \sum_{i=1}^{3} [\![F(x)]\!]_i = \sum_{i=1}^{3} \left(2[\![x^3]\!]_i + 4x_i\right) \pmod 5$$
$$= 0 + 2 + 4 = 6 \equiv 1 \pmod 5.$$

This demonstrates that the majority vote polynomial $F(x)$ can be securely computed via Beaver triples without revealing any individual user's input, while producing an output equivalent to that of the standard non-secure majority voting protocol.

## D   PRECOMPUTED TABLE OF MAJORITY VOTE POLYNOMIALS $F(\mathbf{x})$

The majority vote polynomial $F(\mathbf{x})$ can be efficiently precomputed once the number of users $n$ and the tie-breaking policy are determined in the offline phase. Specifically, for each given $n$ and the tie-breaking rule, the corresponding polynomial can be systematically derived using Eq. 1.

Table 4 presents representative examples of precomputed polynomials according to tie-breaking policies.

Table 4: Precomputed majority vote polynomials $F(\mathbf{x})$ according to tie-breaking policies

| #Users | $\text{sign}(0) \in \{-1, +1\}$ | $\text{sign}(0) = 0$ |
|---|---|---|
| $n = 2$ | $\mathbf{x}^2 + 2\mathbf{x} + 2 \pmod 3$ | $2\mathbf{x} \pmod 3$ |
| $n = 3$ | $2\mathbf{x}^3 + 4\mathbf{x} \pmod 5$ | $2\mathbf{x}^3 + 4\mathbf{x} \pmod 5$ |
| $n = 4$ | $\mathbf{x}^4 + 3\mathbf{x}^3 + \mathbf{x} + 4 \pmod 5$ | $3\mathbf{x}^3 + \mathbf{x} \pmod 5$ |
| $n = 5$ | $3\mathbf{x}^5 + 2\mathbf{x}^3 + 3\mathbf{x} \pmod 7$ | $3\mathbf{x}^5 + 2\mathbf{x}^3 + 3\mathbf{x} \pmod 7$ |
| $n = 6$ | $\mathbf{x}^6 + 4\mathbf{x}^5 + 5\mathbf{x}^3 + 4\mathbf{x} + 6 \pmod 7$ | $4\mathbf{x}^5 + 5\mathbf{x}^3 + 4\mathbf{x} \pmod 7$ |

## E   PROOF OF THEOREM 1

We analyze the convergence of the proposed SIGNSGD algorithm with subgroup-based majority vote. The analysis builds upon standard assumptions from stochastic optimization, and extends the convergence result in (Bernstein et al., 2018a) to a hierarchical subgrouping framework.

### E.1   ASSUMPTIONS

**Assumption 1 (Lower bound).** For all $\boldsymbol{\theta}$, there exists a constant $f^\star$ such that $f(\boldsymbol{\theta}) \geq f^\star$.

**Assumption 2 ($L$-Smoothness).** Let $\nabla f(\boldsymbol{\theta})$ denote the gradient of the objective $f(\cdot)$ at $\boldsymbol{\theta}$. Then for all $\boldsymbol{\theta}, \boldsymbol{\phi} \in \mathbb{R}^d$, we have:

$$|f(\boldsymbol{\phi}) - f(\boldsymbol{\theta}) - \nabla f(\boldsymbol{\theta})^T(\boldsymbol{\phi} - \boldsymbol{\theta})| \leq \frac{1}{2}\sum_{i=1}^d L_i(\boldsymbol{\phi}_i - \boldsymbol{\theta}_i)^2,$$

for some non-negative vector $\vec{L} := [L_1, \ldots, L_d]$.

**Assumption 3 (Variance bound).** Given $\boldsymbol{\theta} \in \mathbb{R}^d$, the stochastic gradient oracle returns an unbiased estimate $\nabla \tilde{f}(\boldsymbol{\theta})$ such that:

$$\mathbb{E}[\nabla \tilde{f}(\boldsymbol{\theta})] = \nabla f(\boldsymbol{\theta}), \quad \mathbb{E}[(\nabla \tilde{f}_i(\boldsymbol{\theta}) - \nabla f_i(\boldsymbol{\theta}))^2] \leq \sigma_i^2,$$

for a vector of non-negative constants $\vec{\sigma} := [\sigma_1, \ldots, \sigma_d]$.

**Assumption 4 (Unimodal, symmetric noise).** Each coordinate of the stochastic gradient $\nabla \tilde{f}(\boldsymbol{\theta})$ has a symmetric and unimodal distribution centered at the true gradient component.

### E.2 PROOF OF CONVERGENCE FOR THE PROPOSED HI-SAFE FRAMEWORK

*Proof.* We first recall the original convergence guarantee for SIGNSGD-MV.

**Theorem 3** (Convergence of SIGNSGD-MV (Bernstein et al., 2018a))**.** *Run Algorithm 2 for $K$ iterations with learning rate $\eta = 1/\sqrt{K\|\vec{L}\|_1}$ and mini-batch size $m_k = K$. Let $N_t = K^2$ be the total number of stochastic gradient evaluations per user. Then the algorithm satisfies the following convergence guarantee:*

$$\mathbb{E}\left[\frac{1}{K}\sum_{k=0}^{K-1}\|\mathbf{g}_k\|_1\right]^2 \leq \frac{1}{\sqrt{N_t}}\left(\sqrt{\|\vec{L}\|_1}(f_0 - f^\star + \tfrac{1}{2}) + \frac{2}{\sqrt{n}}\|\vec{\sigma}\|_1\right)^2.$$

We now extend this result to the hierarchical majority vote with subgrouping algorithm.

We extend Theorem 3 to the case where the $n$ users are partitioned into $\ell$ subgroups $\mathcal{G}_1, \ldots, \mathcal{G}_\ell$ of equal size $n_1 = n/\ell$.

At each iteration $k$, each user $i \in \mathcal{G}_j$ computes a stochastic gradient $\nabla \tilde{f}^{(i)}(\boldsymbol{\theta}_k)$, and the subgroup $\mathcal{G}_j$ computes the coordinate-wise majority vote:

$$\hat{\mathbf{g}}_k^{(j)} := \text{sign}\left(\sum_{i \in \mathcal{G}_j}\text{sign}\left(\nabla \tilde{f}^{(i)}(\boldsymbol{\theta}_k)\right)\right) \in \{\pm 1\}^d.$$

The server then computes the global update direction as:

$$\hat{\mathbf{g}}_k = \text{sign}\left(\sum_{j=1}^\ell \hat{\mathbf{g}}_k^{(j)}\right).$$

**Error Probability per User and Subgroup.** From Assumption 4 and (Bernstein et al., 2018a, Lemma D.1), the sign of a single user's coordinate is incorrect with probability at most:

$$\mathbb{P}\left[\text{sign}(\nabla \tilde{f}_j^{(i)}(\boldsymbol{\theta}_k)) \neq \text{sign}(\nabla f_j(\boldsymbol{\theta}_k))\right] \leq \exp\left(-\frac{g_{k,j}^2}{2\sigma_j^2}\right).$$

Using Hoeffding-type concentration for the sum of $n_1$ independent user signs in subgroup $\mathcal{G}_j$, the probability that the majority vote in coordinate $j$ within subgroup $\mathcal{G}_j$ is incorrect satisfies:

$$\mathbb{P}\left[\hat{\mathbf{g}}_{k,j}^{(j)} \neq \text{sign}(\nabla f_j(\boldsymbol{\theta}_k))\right] \leq \exp\left(-\frac{g_{k,j}^2 n_1}{2\sigma_j^2}\right).$$

**Global Majority over Subgroups.** Since each $\hat{\mathbf{g}}_k^{(j)}$ is obtained independently from its subgroup, the aggregated sign vector $\hat{\mathbf{g}}_k$ reflects the majority of $\ell$ independent majority-vote sign vectors.

We now incorporate two sources of errors:

- Subgroup-level (intra-subgroup) aggregation noise, scaling as $1/\sqrt{n_1}$ per subgroup.

- Global (inter-subgroup) aggregation noise across $\ell$ subgroups, which can be expressed asymptotically as a function of $\ell$.

Following the standard analysis in (Bernstein et al., 2018a), the per-coordinate error probability at the subgroup level leads to a variance term proportional to:

$$\frac{2}{\sqrt{n_1}}\|\vec{\sigma}\|_1.$$

Additionally, the global aggregation error, where each subgroup's output is assumed to be correct with probability $q \in (0.5, 1]$, is given by:

$$\mathbb{P}(\text{global error}) \approx \Phi\left(-\sqrt{\ell} \cdot \alpha_q\right) \approx \frac{1}{\sqrt{2\pi\ell} \cdot \alpha_q} \cdot \exp\left(-\frac{\ell\alpha_q^2}{2}\right), \qquad (11)$$

where

$$\alpha_q := \frac{(2q-1)}{2\sqrt{q(1-q)}} > 0.$$

Accordingly, the global aggregation noise term in the convergence bound behaves as:

$$c' \cdot \sqrt{\mathbb{P}(\text{global error})} = \mathcal{O}\left(\ell^{-1/4} \cdot \exp\left(-\frac{\ell\alpha_q^2}{4}\right)\right).$$

**Concluding the Bound.** Combining the intra- and inter-subgroup error terms, we obtain the following convergence guarantee:

$$\mathbb{E}\left[\frac{1}{K}\sum_{k=0}^{K-1}\|\mathbf{g}_k\|_1\right]^2 \leq \frac{1}{\sqrt{N_t}}\left(\sqrt{\|\vec{L}\|_1}(f_0 - f^\star + \tfrac{1}{2}) + \frac{2}{\sqrt{n_1}}\|\vec{\sigma}\|_1 + c' \cdot \ell^{-1/4} \cdot \exp\left(-\frac{\ell\alpha_q^2}{4}\right)\right)^2,$$

where $c' > 0$ is a constant reflecting the global aggregation perturbation scale.

This completes the proof. $\qquad\qquad\square$

# F  SECURITY PROOF OF THEOREM 2

## F.1  PRELIMINARIES AND NOTATION

Let $\mathbb{F}_{p_1}$ and $\mathbb{F}_{p_2}$ be prime fields with $p_1 > n_1$ and $p_2 > \ell$, respectively. For each subgroup $\mathcal{G}_j$ of size $n_1$, define the subgroup aggregate $\mathbf{x}_j := \sum_{i=1}^{n_1} \mathbf{x}_{i,j}$ and its majority $\mathbf{s}_j := \text{sign}(\mathbf{x}_j) \in \{-1, 0, +1\}$ (tie-breaking via $\tau$). Let $F(\cdot)$ denote the finite-field majority polynomial (constructed via FLT) so that $F(\mathbf{x}_j) = \mathbf{s}_j \pmod{p_1}$ and, at the inter-group layer, the analogous polynomial produces the final majority $\mathbf{s} = \text{sign}(\sum_{j=1}^{\ell} \mathbf{s}_j) \in \{-1, 0, +1\}$.

We assume additive secret sharing and Beaver triples $(a, b, c)$ with $c = a \cdot b$, generated via MPC among users, as the basis for secure multiplications. For a secret value $\mathbf{z}$, the parties hold shares $\{[\![\mathbf{z}]\!]_i\}_{i \in S}$ with $\sum_{i \in S}[\![\mathbf{z}]\!]_i = \mathbf{z}$. Masked openings are computed as

$$\boldsymbol{\delta} = \mathbf{x} - \mathbf{a}, \qquad \boldsymbol{\epsilon} = \mathbf{y} - \mathbf{b},$$

which are publicly revealed as sums of masked share-differences.

Throughout, we consider a semi-honest adversary $\mathcal{A}$ corrupting at most $t \leq n - 1$ users globally, and *assume every subgroup contains at least one honest user*, i.e., $t_j \leq n_1 - 1$ in each $\mathcal{G}_j$.

## F.2  LEMMA A: PRIVACY OF BEAVER MASKED-OPENINGS (LOCAL)

**Claim.** Consider a single multiplication in subgroup $\mathcal{G}_j$ under additive sharing over $\mathbb{F}_{p_1}$. If at most $t_j \leq n_1 - 1$ users are corrupted in $\mathcal{G}_j$, then the publicly opened masked differences $(\boldsymbol{\delta}, \boldsymbol{\epsilon}) = (\mathbf{x} - \mathbf{a}, \mathbf{y} - \mathbf{b})$ are (computationally) indistinguishable from uniform over $\mathbb{F}_{p_1}^2$ and independent of the true inputs $(\mathbf{x}, \mathbf{y})$.

*Proof.* Let the parties hold additive shares of $\mathbf{x}, \mathbf{y}$ and of the Beaver triple $(\mathbf{a}, \mathbf{b}, \mathbf{c})$ with $\mathbf{c} = \mathbf{a} \cdot \mathbf{b}$, where $\mathbf{a}, \mathbf{b} \leftarrow_R \mathbb{F}_{p_1}$ are sampled independently of $(\mathbf{x}, \mathbf{y})$ in preprocessing. With $t_j \leq n_1 - 1$, there exists at least one honest party $h \in \mathcal{G}_j$. From $\mathcal{A}$'s view, the contributions of $h$'s unknown mask-shares $(a_h, b_h)$ make $(\mathbf{x}_h - a_h, \mathbf{y}_h - b_h)$ uniformly random in $\mathbb{F}_{p_1}^2$, independent of $(\mathbf{x}, \mathbf{y})$. Since $(\boldsymbol{\delta}, \boldsymbol{\epsilon})$ are sums of per-party masked differences, they equal a fixed (adversary-known) offset plus an independent uniform pair, hence are uniform and input-independent. Thus a simulator can sample $(\hat{\boldsymbol{\delta}}, \hat{\boldsymbol{\epsilon}}) \leftarrow_R \mathbb{F}_{p_1}^2$ to reproduce indistinguishable openings. $\qquad\square$

## F.3  LEMMA B: PRIVACY OF INTRA-SUBGROUP EVALUATION WITHOUT PLAINTEXT $\mathbf{s}_j$

**Claim.** In subgroup $\mathcal{G}_j$, suppose $F(\mathbf{x}_j)$ is evaluated via an arithmetic circuit using Beaver-based multiplications over $\mathbb{F}_{p_1}$ and *the output remains secret-shared* (no plaintext reconstruction of $\mathbf{s}_j$). Under Lemma F.2, the entire intra-subgroup transcript is simulatable *without knowing $\mathbf{s}_j$ in plaintext*. In particular, there exists a simulator that outputs a view indistinguishable from the real one while producing secret shares that add up to some dummy value $\hat{\mathbf{s}}_j \in \{-1, 0, +1\}$, which is never revealed.

*Proof.* The circuit for $F(\mathbf{x}_j)$ consists of additions on shares (perfectly private) and Beaver-based multiplications. By Lemma F.2, for each multiplication gate $g$, the opened masks $(\boldsymbol{\delta}_g, \boldsymbol{\epsilon}_g)$ are uniform and input-independent, hence simulatable by random draws in $\mathbb{F}_{p_1}^2$. Since the subgroup output is not reconstructed, the simulator may fix an arbitrary dummy outcome $\hat{\mathbf{s}}_j \in \{-1, 0, +1\}$ and secret-share it by picking $\{[\![\hat{\mathbf{s}}_j]\!]_i\}_{i \in \mathcal{G}_j}$ uniformly at random subject to $\sum_i [\![\hat{\mathbf{s}}_j]\!]_i = \hat{\mathbf{s}}_j$. It then generates per-gate masked openings uniformly at random and updates local (simulated) shares using the Beaver correctness relation to be consistent with the chosen $\{[\![\hat{\mathbf{s}}_j]\!]_i\}$. Because no plaintext $\mathbf{s}_j$ is ever revealed and all public openings are input-independent uniforms, the resulting transcript is indistinguishable from the real execution. $\qquad\square$

## F.4  LEMMA C: PRIVACY OF INTER-SUBGROUP COMPOSITION GIVEN ONLY $\mathbf{s}$

**Claim.** Assume each subgroup $\mathcal{G}_j$ produces a secret sharing of its (not-opened) output $F(\mathbf{x}_j)$ as in Lemma F.3. At the inter-group stage over $\mathbb{F}_{p_2}$, where the final plaintext majority $\mathbf{s}$ is revealed, the entire transcript is simulatable *given only* $\mathbf{s}$.

*Proof.* The inter-group computation takes as inputs the secret shares $\{[\![F(\mathbf{x}_j)]\!]_i\}_i$ arriving from all subgroups (no plaintext $\mathbf{s}_j$ is available to the server/adversary). In simulation, we first *choose dummy subgroup outcomes* $\{\hat{\mathbf{s}}_j\}_{j=1}^{\ell} \in \{-1, 0, +1\}^{\ell}$ such that

$$\mathrm{sign}\Big( \sum_{j=1}^{\ell} \hat{\mathbf{s}}_j \Big) = \mathbf{s}.$$

(If $\mathbf{s} = +1$, pick any vector with a strict positive sum, etc.) We then secret-share each $\hat{\mathbf{s}}_j$ among parties (including corrupted ones) by sampling shares uniformly at random conditioned on summing to $\hat{\mathbf{s}}_j$. Next, we simulate every Beaver-based multiplication gate at the inter-group layer by sampling masked openings $(\boldsymbol{\delta}_g, \boldsymbol{\epsilon}_g) \leftarrow_R \mathbb{F}_{p_2}^2$ independently and updating shares according to the Beaver algebra so that the final reconstruction equals the required plaintext $\mathbf{s}$. This is always possible since (i) masked openings are uniform (input-independent) by the same argument as Lemma F.2 (now over $\mathbb{F}_{p_2}$), and (ii) we control the dummy inputs $\{\hat{\mathbf{s}}_j\}$ and their shares. Therefore, the inter-group transcript (including all public masked openings and the final revealed $\mathbf{s}$) is indistinguishable from the real one given only $\mathbf{s}$. $\qquad\square$

## F.5 PROOF OF THEOREM 2

*Proof. Simulator construction (explicit steps).* Given the corrupted inputs $\{\mathbf{x}_{i,j}\}_{i\in\mathcal{C}}$ and the final plaintext majority $\mathbf{s}$:

1. **Preprocessing (both layers).** Sample all Beaver triples $(\mathbf{a}, \mathbf{b}, \mathbf{c})$ gate-wise with $\mathbf{a}, \mathbf{b} \leftarrow_R$ uniform, and secret-share them to parties (including corrupted ones) as in the real protocol.

2. **Intra-subgroup stage.** For each subgroup $\mathcal{G}_j$:

   (a) Pick a dummy $\hat{\mathbf{s}}_j \in \{-1, 0, +1\}$ arbitrarily (independent of real $\mathbf{s}_j$).
   
   (b) Secret-share $\hat{\mathbf{s}}_j$ among the $n_1$ users by sampling $\{[\![\hat{\mathbf{s}}_j]\!]_i\}_{i\in\mathcal{G}_j}$ uniformly at random subject to summing to $\hat{\mathbf{s}}_j$.
   
   (c) For each multiplication gate, sample $(\hat{\boldsymbol{\delta}}_g, \hat{\boldsymbol{\epsilon}}_g) \leftarrow_R \mathbb{F}_{p_1}^2$ and produce the corresponding public openings; update simulated shares via Beaver's correctness relation so that the final (unopened) output shares equal $\{[\![\hat{\mathbf{s}}_j]\!]_i\}$.

3. **Inter-group stage.** Choose the vector $\{\hat{\mathbf{s}}_j\}_{j=1}^{\ell}$ such that $\text{sign}(\sum_j \hat{\mathbf{s}}_j) = \mathbf{s}$ (override any previous arbitrary choices if necessary). Using the existing secret shares of $\hat{\mathbf{s}}_j$, simulate the inter-group circuit: for each multiplication gate over $\mathbb{F}_{p_2}$, sample $(\hat{\boldsymbol{\delta}}_g, \hat{\boldsymbol{\epsilon}}_g) \leftarrow_R \mathbb{F}_{p_2}^2$, update shares accordingly, and finally reconstruct the plaintext output $\mathbf{s}$.

*Indistinguishability.* By Lemma F.2, all public masked openings at both layers are input-independent uniforms; thus replacing them by uniform samples preserves distribution. By Lemma F.3, since $\mathbf{s}_j$ are never opened, substituting dummy (secret) outputs $\hat{\mathbf{s}}_j$ yields an indistinguishable view. By Lemma F.4, the inter-group transcript is simulatable given only $\mathbf{s}$. Therefore,

$$\text{REAL}_{\Pi,\mathcal{A}}\big(\{\mathbf{x}_{i,j}\}_{i\in\mathcal{C}}\big) \ \approx_c \ \text{SIM}_{\mathcal{A}}\big(\{\mathbf{x}_{i,j}\}_{i\in\mathcal{C}}, \mathbf{s}\big).$$

$\square$

## F.6 COROLLARY: END-TO-END PRIVACY (SHARE-$\mathbf{s}_j$)

Under the assumptions above, Hi-SAFE achieves end-to-end privacy in the Share-$\mathbf{s}_j$ setting:

$$\text{REAL}_{\Pi} \ \approx_c \ \text{SIM}_{\Pi}, \qquad \text{leak} = \{\mathbf{s}\}.$$

No adversary corrupting at most $t \leq n-1$ users can distinguish the real transcript from the simulated one, beyond the final $\mathbf{s}$.

## F.7 CORRUPTION TOLERANCE OF HI-SAFE

Based on Theorem 2, we derive the maximum *privacy threshold* of Hi-SAFE under the semi-honest security model.

**Corollary 3.1** (Privacy Corruption Tolerance of Hi-SAFE). *Consider a federated learning system with $n$ users applying the Hi-SAFE protocol.*

- *Flat Majority Vote (Non-Subgrouping): Hi-SAFE preserves privacy against any coalition of*

$$t \ \leq \ n-1$$

*corrupted users, revealing only the final majority vote $F(\mathbf{x})$.*

- *Hierarchical Majority Vote (Subgrouping): Partition the $n$ users into $\ell$ subgroups, each of size $n_1 = n/\ell$. If every subgroup contains at least one honest user (i.e., $t_j \leq n_1 - 1$ for all $j$), then Hi-SAFE preserves privacy against*

$$t \ \leq \ n-1$$

*corrupted users in total, revealing only the global majority $F(\mathbf{x})$.*

*Proof Sketch.* In the flat case, additive $n$-out-of-$n$ secret sharing ensures that even if $n-1$ shares are known, the remaining share perfectly hides the honest input. The MPC-based Beaver triple generation guarantees that the masked openings $(\boldsymbol{\delta}, \boldsymbol{\epsilon})$ are input-independent. Thus, privacy holds up to $t = n - 1$ corruptions.

In the hierarchical case, each subgroup performs local aggregation securely. Compromising all $n_1$ users of one subgroup reveals no additional information about honest inputs outside that group. Provided that every subgroup contains at least one honest user, the end-to-end privacy proof (Theorem 2) extends directly, and the overall system preserves privacy against up to $t = n - 1$ corrupted users. □

### F.7.1 CORRUPTION TOLERANCE AND SUBGROUPING

Table 5: Summary of corruption tolerance

| Scenario | Tolerance Threshold |
|---|---|
| Flat Majority Vote (Non-Subgrouping) | $t \le n - 1$ |
| Hierarchical Majority Vote (Subgrouping) | $t \le n - 1$ |

Flat majority vote aggregation allows for a maximum corruption tolerance of up to $t = n-1$, but incurs a per-user communication cost of $\mathcal{O}(\log^2 p)$. In contrast, hierarchical majority vote aggregation reduces the per-user communication complexity to $\mathcal{O}(\log^2 p_1)$, where $p_1 \ll p$, while maintaining the same maximum corruption tolerance of $t = n - 1$ (see Table 6 in Appendix G). Therefore, the hierarchical aggregation can be a good choice when network scalability is a critical requirement.

## G TIE-BREAKING POLICIES AND EFFECT OF HIERARCHICAL AGGREGATION

### G.1 TIE-BREAKING POLICIES IN MAJORITY VOTE ENCRYPTION

In the proposed subgroup-based FL framework, the tie-breaking policy adopted in the majority voting process significantly influences both computational efficiency and communication cost. We distinguish two levels of majority vote—*intra-subgroup* and *inter-subgroup*—each of which may apply different tie-breaking rules.

**Intra-Subgroup Majority Vote:**

- **Case A:** $\text{sign}(0) \in \{-1, 1\}$ (tie deterministically mapped to binary decision (requiring only 1-bit precision))
- **Case B:** $\text{sign}(0) = 0$ (tie represented as third state (requiring 2-bit precision))

*Note that Case B increases the computational resolution for intra-subgroup aggregation, but incurs no additional uplink communication cost since these computations are entirely internal to the server.*

**Inter-Subgroup Majority Vote:**

- **Case 1:** $\text{sign}(0) \in \{-1, 1\}$ (tie mapped to binary decision (resulting in a 1-bit downlink))
- Case 2: $\text{sign}(0) = 0$ (tie represented as third state (increasing the downlink to 2 bits))

**Combined Tie-Breaking Configurations:** We consider the following four configurations that combine the intra- and inter-subgroup tie-breaking policies:

- **Case A-1 (1-bit tie-breaking):** 1-bit Intra/1-bit Inter (most efficient)
- **Case B-1 (2-bit tie-breaking):** 2-bit Intra/1-bit Inter (higher resolution within subgroup aggregation without increasing communication cost)
- Case A-2: 1-bit Intra/2-bit Inter (higher resolution in global aggregation, but increasing communication cost)
- Case B-2: 2-bit Intra/2-bit Inter (maximum resolution, but increasing communication cost)

**Remark:** Configurations involving Case 2 (2-bit downlink) are incompatible with the SIGNSGD-MV protocol considered in this work, which strictly assumes a 1-bit communication model for both uplink and downlink. Accordingly, our analysis primarily focuses on the tie-breaking policies within the 1-bit communication constraint regime, i.e., Cases A-1 and B-1.

### G.1.1 Effect of Hierarchical Aggregation

To reduce the overhead of secure multiparty computation, the proposed *hierarchical aggregation* technique partitions the total of $n$ users into $\ell$ disjoint subgroups, each of size $n_1 = n/\ell$. Within each subgroup, secure aggregation is performed using a smaller prime modulus $p_1$, which significantly reduces the number of required Beaver multiplication subrounds and the corresponding communication overhead:

$$\text{\# Secure multiplication subrounds} = \mathcal{O}(\lceil \log p_1 - 1 \rceil), \quad \text{Communication cost} = \mathcal{O}(\log^2 p_1).$$

Although the total communication cost across all subgroups becomes $\mathcal{O}(\ell \cdot \log^2 p_1)$, it remains asymptotically lower than that of the flat aggregation scheme, which requires $\mathcal{O}(\log^2 p)$ when secure aggregation is performed over all $n$ users without subgrouping. This gap widens as the number of users $n$ increases, since the majority vote polynomial $F(\mathbf{x})$ becomes more complex, resulting in a larger number of multiplication terms and hence higher communication and computation overhead under flat aggregation.

Table 6 summarizes the key differences between the flat and hierarchical aggregation schemes. The proposed hierarchical framework achieves scalable and bandwidth-efficient secure aggregation, especially under large-scale FL settings. Notably, when the number of subgroups $\ell$ is optimally selected, the per-user communication cost approaches that of the baseline SIGNSGD-MV protocol. Therefore, this framework is well-suited for resource-constrained FL systems that require both high privacy guarantees and low communication overhead.

Table 6: Comparison between flat and hierarchical aggregation schemes

| Metric | Flat Aggregation | Hierarchical Aggregation (Optimal $\ell^\star$) |
|:---:|:---:|:---:|
| # Users | $n$ | $n_1 = n/\ell \ll n$ |
| Prime Modulus | $p\ (> n)$ | $p_1 \ll p$ |
| Latency | $\lceil \log p - 1 \rceil$ | $\lceil \log p_1 - 1 \rceil \approx 2$ |
| Per-User Comm. Cost | $\mathcal{O}(\log^2 p)$ | $\mathcal{O}(\log^2 p_1) \approx \mathcal{O}(1)$ |
| Total Comm. Cost | $\mathcal{O}(\log^2 p)$ | $\mathcal{O}(\ell \cdot \log^2 p_1) \approx \mathcal{O}(\ell)$ |
| Corruption Tolerance | $t \leq n - 1$ | $t \leq n - 1$ |

## H Computation and Runtime Overhead in Secure Evaluation

To evaluate the efficiency of the proposed secure evaluation scheme (Algorithm 1), we measured the offline preprocessing cost, corresponding to Beaver triple generation, as well as the online execution cost for secure polynomial evaluation under optimal subgrouping. The results are summarized in Table 7.

Table 7: Runtime and computational complexity of Algorithm 1 under practical FL settings.

| Phase | Operation | Complexity | Average Runtime (sec) |
|:---|:---:|:---:|:---:|
| Offline | Beaver triple generation | $O(Rn)$ | $< 0.01$ |
| Offline | Precomputation of $F(\mathbf{x})$ | $\mathcal{O}(n_1 \cdot \log p_1)$ | $< 0.01$ |
| Online | Secure evaluation of $F(\mathbf{x})$ | $\mathcal{O}(Rn + \deg(F_{\text{sub}}))$ | $0.01-0.02$ |
| **Total** | (Offline + Online) | $\mathcal{O}(Rn + \deg(F_{\text{sub}}))$ | $< 0.03$ |

As shown in Table 7, the total runtime of Algorithm 1 consistently remains below 0.03 seconds on average, encompassing both Beaver triple generation and polynomial evaluation. This compu-

tational cost is negligible when compared with that of Algorithm 2 (secure aggregation with local training and sign processing), which typically incurs more than 10 seconds per global round in FL scenarios such as FMNIST under non-IID settings. Consequently, the overhead introduced by Algorithm 1 is unlikely to constitute a performance bottleneck in practical deployments, even in large-scale FL environments. In particular, the runtime difference between Algorithm 1 and Algorithm 2 highlights the advantage of employing a lightweight cryptographic aggregation strategy, as the secure evaluation component adds only a marginal cost relative to the overall training process. Furthermore, it is important to note that no low-level optimizations were incorporated into the current implementation; hence, the reported runtime should be regarded as a conservative baseline. With carefully engineered software or hardware acceleration (e.g., vectorized operations, parallelization on GPUs, or dedicated secure computation libraries), the runtime can be further reduced, thereby reinforcing the practicality of Algorithm 1 for real-world FL applications.

## H.1 COMPUTATIONAL COMPLEXITY ANALYSIS OF MAJORITY VOTE POLYNOMIAL

We analyze the computational complexity of evaluating the majority vote polynomial $F(\mathbf{x})$, which plays a central role in our secure aggregation scheme. Let $\mathbf{m} = \sum_i \mathbf{m}_i$ takes only $n + 1$ distinct values from $\{-n, -n+2, \ldots, n\}$.

$$F(\mathbf{x}) = \sum_{\mathbf{m}\in\{-n,-n+2,\ldots,n-2,n\}} \text{sign}(\mathbf{m}) \cdot \left[ 1 - (\mathbf{x} - \mathbf{m})^{p-1} \right] \pmod{p},$$

This is the number of terms to $\mathcal{O}(n)$, and per-term cost becomes:

- $\text{sign}(\mathbf{m})$: $\mathcal{O}(1)$
- $(\mathbf{x} - \mathbf{m})^{p-1}$: $\mathcal{O}(\log p)$

$$\text{Naive complexity:} \quad \boxed{\mathcal{O}(n \cdot \log p)}$$

**Further Reduction via Subgrouping.** To further reduce computational overhead, our protocol employs a subgrouping mechanism where users are partitioned into subgroups of size $n_1 \ll n$. The majority vote polynomial is locally constructed within each subgroup. This results in a much smaller polynomial degree and field size ($p_1 \ll p$), and complexity becomes:

$$\text{With subgrouping:} \quad \boxed{\mathcal{O}(n_1 \cdot \log p_1)}.$$

Moreover, this polynomial is generated once in the offline phase and reused across rounds, incurring no additional cost during the online aggregation.

Table 8: Comparison of Computational Complexity

| Method | # Terms | Total Complexity |
|---|---|---|
| Naive Enumeration | $\leq n + 1$ | $\mathcal{O}(n \cdot \log p)$ |
| With Subgrouping | $\leq n_1 + 1$ | $\mathcal{O}(n_1 \cdot \log p_1)$ |

**Complexity Comparison.** These optimizations enable efficient and scalable polynomial evaluation, making our approach practical for large-scale FL deployments.

## I ADDITIONAL EXPERIMENT RESULTS

### I.1 FURTHER DETAILS ABOUT EXPERIMENT SET-UP

**Experimental Settings.** We validate Hi-SAFE on MNIST (LeCun et al., 1998), FMNIST (Xiao et al., 2017), and CIFAR-10 (Krizhevsky et al., 2009) datasets, in the presence of adversaries. The datasets are divided as follows: 60,000 training and 10,000 testing samples for both MNIST and FMNIST, and 50,000 training and 10,000 testing samples for CIFAR-10. We consider $N = 100$

users, each having the same number of data samples. As described in (McMahan et al., 2017), two classes are randomly assigned to each user to induce non-IID situations. Batch normalization layers were omitted during the training of CIFAR-10. At each global round, a fraction $C$ of users is randomly selected to participate. We set $C$ between 0.12 and 0.36 at each global round. Additionally, the details of the hyperparameters for Hi-SAFE are shown in Table 9.

Table 9: Hyperparameters for Hi-SAFE

| Method | Dataset | $\eta$ (learning rate) | Batch Size | Local Epoch |
|---|---|---|---|---|
| Non-Subgrouping | MNIST | 0.001 | 100 | 1 |
| | FMNIST | 0.005 | | |
| | CIFAR-10 | 0.0001 | | |
| Subgrouping | MNIST | 0.001 | 100 | 1 |
| | FMNIST | 0.005 | | |
| | CIFAR-10 | 0.0001 | | |

## I.2 ADDITIONAL EXPERIMENT RESULTS

To validate the generality of our findings, we conducted extensive experiments under various federated learning environments. Specifically, we evaluated the proposed hierarchical aggregation framework across different datasets (FMNIST, MNIST, and CIFAR-10), data distributions (IID and non-IID), and user scales ranging from $n = 12$ to $n = 36$, as illustrated in Figures 4–9.

For each configuration, we compared model performance under 1-bit ($\text{sign}(0) \in \{-1, +1\}$) and 2-bit ($\text{sign}(0) = 0$) tie-breaking, using both non-subgrouping and optimal subgrouping strategies. In the non-subgrouping case, all users are aggregated as a single group, effectively treating the entire population as one subgroup. As a result, inter-subgroup aggregation becomes unnecessary.

The results consistently show that the proposed subgrouping strategy maintains model accuracy comparable to the flat (non-subgrouping) setting while significantly reducing the cost of secure computation. Under the 1-bit tie-breaking setting, the non-subgrouping configuration of Hi-SAFE is functionally equivalent to naive SIGNSGD-MV, except for the added privacy guarantees. Accordingly, we observe that Hi-SAFE achieves performance comparable to existing methods in this setting. In contrast, applying 2-bit tie-breaking improves computational precision on the server side and leads to a slight improvement in model accuracy, with the performance gains being more pronounced under the subgrouping strategy. These results demonstrate that, under the 2-bit tie-breaking setting, Hi-SAFE can outperform conventional methods in terms of both accuracy and privacy preservation.

Moreover, the results demonstrate that 1-bit and 2-bit tie-breaking generally lead to similar accuracy, each exhibiting a trade-off between computational efficiency and numerical precision. While the 1-bit method minimizes computation overhead, the 2-bit strategy improves robustness—particularly in highly heterogeneous or complex tasks such as CIFAR-10, and when the subgroup size $n_1$ is even.

Across all evaluated settings—including both IID and non-IID distributions—the proposed hierarchical scheme demonstrates stable convergence and enhanced communication efficiency, confirming its robustness and adaptability to diverse FL environments.

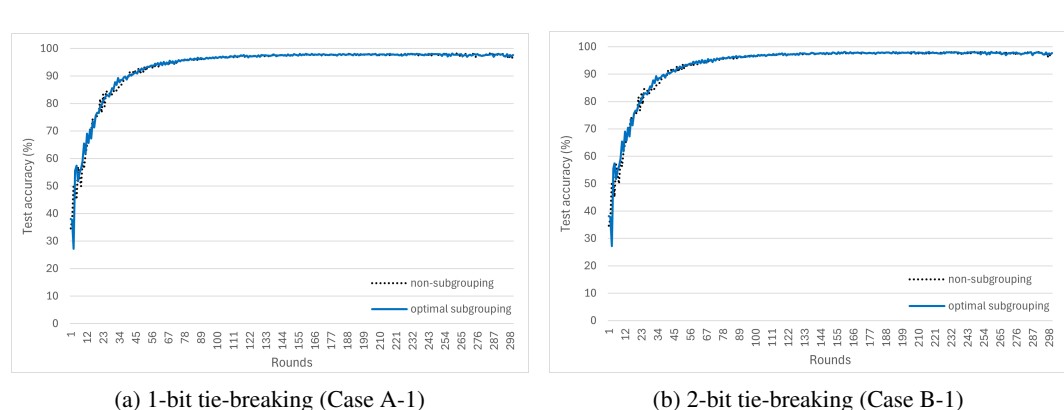

(a) 1-bit tie-breaking (Case A-1)      (b) 2-bit tie-breaking (Case B-1)

Figure 4: Performance comparison of tie-breaking policies on the MNIST dataset under IID setting with $n=12$.

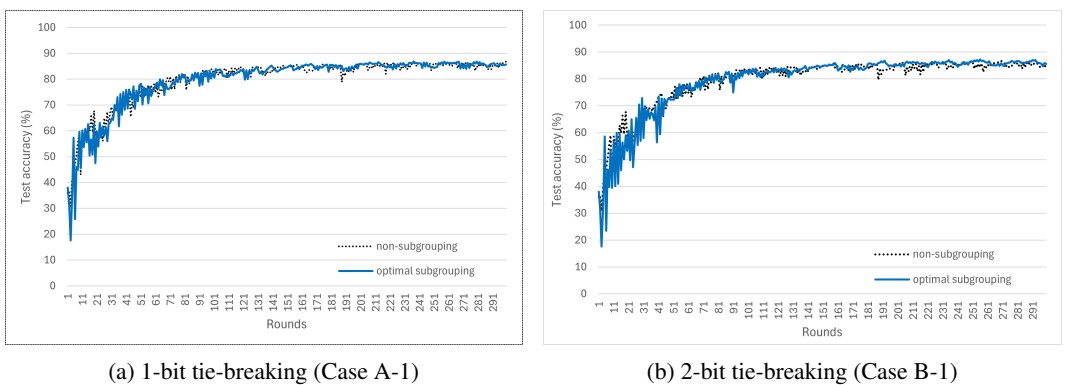

(a) 1-bit tie-breaking (Case A-1)      (b) 2-bit tie-breaking (Case B-1)

Figure 5: Performance comparison of tie-breaking policies on the FMNIST dataset under IID setting with $n=36$.

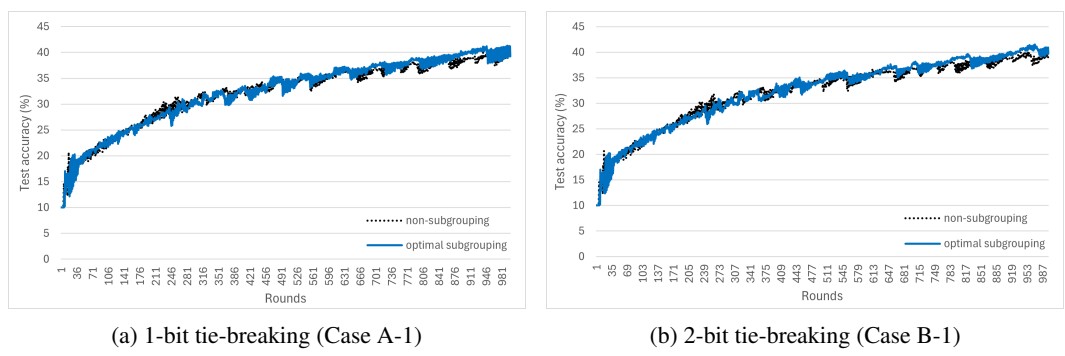

(a) 1-bit tie-breaking (Case A-1)      (b) 2-bit tie-breaking (Case B-1)

Figure 6: Performance comparison of tie-breaking policies on the CIFAR-10 dataset under IID setting with $n=24$.

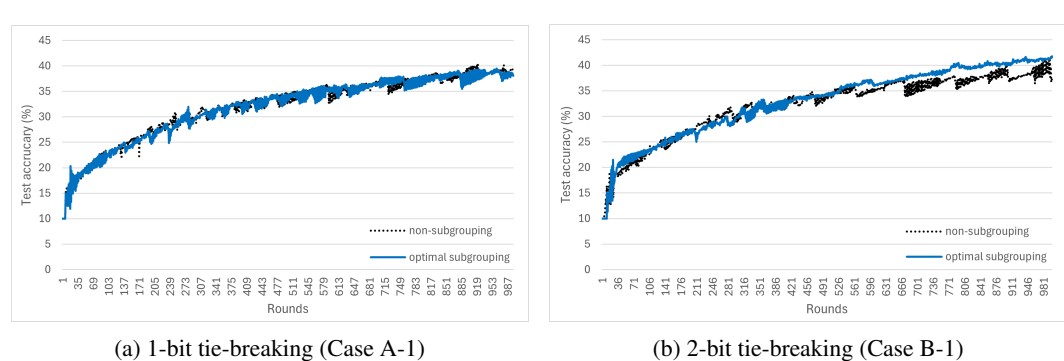

(a) 1-bit tie-breaking (Case A-1)  (b) 2-bit tie-breaking (Case B-1)

Figure 7: Performance comparison of tie-breaking policies on the CIFAR-10 dataset under IID setting with $n = 36$.

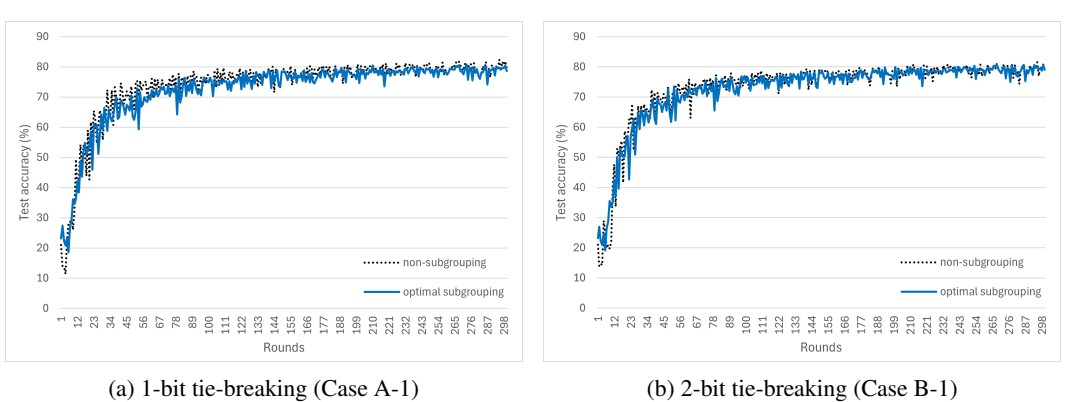

(a) 1-bit tie-breaking (Case A-1)  (b) 2-bit tie-breaking (Case B-1)

Figure 8: Performance comparison of tie-breaking policies on the FMNIST dataset under non-IID setting with $n = 24$.

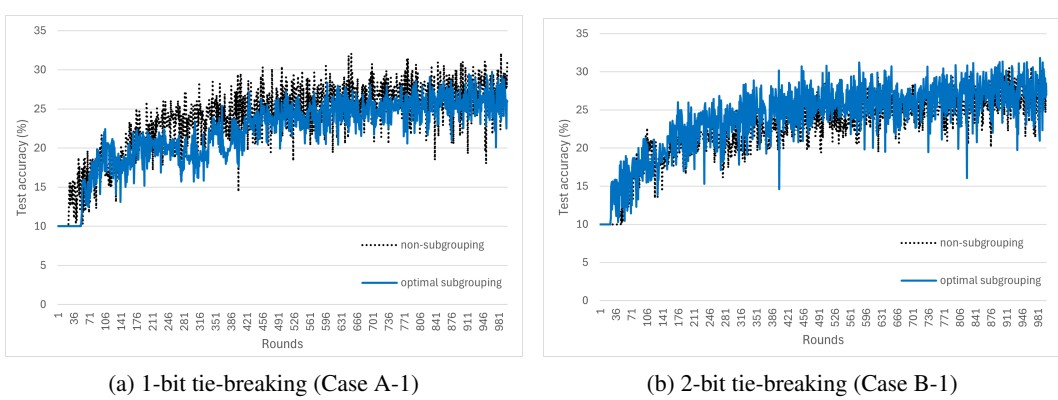

(a) 1-bit tie-breaking (Case A-1)  (b) 2-bit tie-breaking (Case B-1)

Figure 9: Performance comparison of tie-breaking policies on the CIFAR-10 dataset under non-IID setting with $n = 24$.

### I.3 EXTENDED EVALUATION OF THE OPTIMAL SUBGROUPING STRATEGY

To quantify the effect of subgrouping on communication efficiency, we define the total communication cost $C_T$ and the per-user communication cost $C_u$ as follows:

$$C_T = \ell \cdot (R \cdot \lceil \log p_1 \rceil), \quad C_u = R \cdot \lceil \log p_1 \rceil \quad \text{(bits)}, \tag{12}$$

where:

- $\ell = n/n_1$: Number of subgroups (with $\ell = 1$ representing the non-subgrouping case),
- $n$: Total number of users,
- $n_1$: Number of users in each subgroup,
- $p_1$: The smallest prime number strictly greater than $n_1$,
- $\lceil \log p_1 \rceil$: Bit length of the modulus prime $p_1$ used for field representation,
- $\lceil \log p_1 - 1 \rceil$: Number of sequential Beaver subrounds required for secure multiplication (i.e., latency),
- $R$: Total number of secure multiplications, which scales proportionally to $\lceil \log p_1 - 1 \rceil$.

Tables 10 and 11 present a comparison of the per-user communication cost $C_u$ and total cost $C_T$, as well as the associated latency and multiplication cost under various subgroup configurations. Parentheses in the table denote the percentage reduction relative to the baseline case $\ell = 1$, that is, the non-subgrouping case. The results show that choosing an optimal $\ell$ leads to substantial savings not only in total and per-user communication costs but also in the associated latency and multiplication cost, thereby validating the effectiveness of the proposed subgrouping strategy for scalable and communication-efficient secure aggregation in FL.

Table 10: Key metrics across different subgroup configurations

| $n$ | $\ell$ | $n_1$ | $p_1$ | $\lceil \log p_1 \rceil$ | $\lceil \log p_1 - 1 \rceil$ | $R$ | $C_T$ (%) | $C_u$ (%) |
|---|---|---|---|---|---|---|---|---|
| 12 | 1 | 12 | 13 | 4 | 3 | 18 | 72 (-) | 72 (-) |
| 12 | 2 | 6 | 7 | 3 | 2 | 10 | 60 (16.7%) | 30 (58.3%) |
| 12 | 3 | 4 | 5 | 3 | 2 | 6 | 54 (25.0%) | 18 (75.0%) |
| 12 | 4 | 3 | 5 | 3 | 2 | 4 | 48 (33.3%) | 12 (83.3%) |
| 15 | 1 | 15 | 17 | 5 | 4 | 18 | 90 (-) | 90 (-) |
| 15 | 3 | 5 | 7 | 3 | 2 | 8 | 48 (46.7%) | 24 (73.3%) |
| 15 | 5 | 3 | 5 | 3 | 2 | 4 | 60 (33.3%) | 12 (86.7%) |
| 16 | 1 | 16 | 17 | 5 | 4 | 20 | 100 (-) | 100 (-) |
| 16 | 2 | 8 | 11 | 4 | 3 | 14 | 112 (-12.0%) | 56 (44.0%) |
| 16 | 4 | 4 | 5 | 3 | 2 | 6 | 72 (28.0%) | 18 (82.0%) |
| 20 | 1 | 20 | 23 | 5 | 4 | 32 | 160 (-) | 160 (-) |
| 20 | 2 | 10 | 11 | 4 | 3 | 16 | 128 (20.0%) | 64 (60.0%) |
| 20 | 4 | 5 | 7 | 3 | 2 | 8 | 96 (40.0%) | 24 (85.0%) |
| 20 | 5 | 4 | 5 | 3 | 2 | 6 | 90 (43.8%) | 18 (88.7%) |
| 24 | 1 | 24 | 29 | 5 | 4 | 40 | 200 (-) | 200 (-) |
| 24 | 2 | 12 | 13 | 4 | 3 | 18 | 144 (28.0%) | 72 (64.0%) |
| 24 | 3 | 8 | 11 | 4 | 3 | 14 | 168 (16.0%) | 56 (72.0%) |
| 24 | 4 | 6 | 7 | 3 | 2 | 10 | 120 (40.0%) | 30 (85.0%) |
| 24 | 6 | 4 | 7 | 3 | 2 | 6 | 108 (46.0%) | 18 (91.0%) |
| 24 | 8 | 3 | 5 | 3 | 2 | 4 | 96 (52.0%) | 12 (94.0%) |
| 28 | 1 | 28 | 29 | 5 | 4 | 40 | 200 (-) | 200 (-) |
| 28 | 2 | 14 | 17 | 5 | 4 | 22 | 220 (-10.0%) | 110 (45.0%) |
| 28 | 4 | 7 | 11 | 4 | 3 | 14 | 224 (-12.0%) | 56 (72.0%) |
| 28 | 7 | 4 | 5 | 3 | 2 | 6 | 126 (37.0%) | 18 (91.0%) |
| 30 | 1 | 30 | 31 | 5 | 4 | 38 | 190 (-) | 190 (-) |
| 30 | 2 | 15 | 17 | 4 | 3 | 20 | 200 (-5.3%) | 100 (47.4%) |
| 30 | 3 | 10 | 11 | 4 | 3 | 16 | 192 (-1.1%) | 64 (66.3%) |
| 30 | 5 | 6 | 7 | 3 | 2 | 10 | 150 (21.1%) | 30 (84.2%) |
| 30 | 6 | 5 | 7 | 3 | 2 | 8 | 144 (24.2%) | 24 (87.4%) |
| 30 | 10 | 3 | 5 | 3 | 2 | 4 | 120 (36.8%) | 12 (93.7%) |
| 36 | 1 | 36 | 37 | 6 | 5 | 46 | 276 (-) | 276 (-) |
| 36 | 2 | 18 | 19 | 5 | 4 | 26 | 260 (5.8%) | 130 (52.9%) |
| 36 | 3 | 12 | 13 | 4 | 3 | 18 | 216 (21.7%) | 72 (73.9%) |
| 36 | 4 | 9 | 11 | 4 | 3 | 14 | 224 (18.8%) | 56 (79.7%) |
| 36 | 6 | 6 | 7 | 3 | 2 | 10 | 180 (34.8%) | 30 (89.1%) |
| 36 | 9 | 4 | 5 | 3 | 2 | 6 | 162 (41.3%) | 18 (93.5%) |
| 36 | 12 | 3 | 5 | 3 | 2 | 4 | 144 (47.8%) | 12 (95.7%) |
| 40 | 1 | 40 | 41 | 6 | 5 | 48 | 288 (-) | 288 (-) |
| 40 | 2 | 20 | 23 | 5 | 4 | 32 | 320 (-11.1%) | 160 (44.4%) |
| 40 | 4 | 10 | 11 | 4 | 3 | 16 | 256 (11.1%) | 64 (77.8%) |
| 40 | 5 | 8 | 11 | 4 | 3 | 14 | 280 (2.8%) | 56 (80.6%) |
| 40 | 8 | 5 | 7 | 3 | 2 | 8 | 192 (33.3%) | 24 (91.7%) |
| 40 | 10 | 4 | 5 | 3 | 2 | 6 | 180 (37.5%) | 18 (93.8%) |

Table 11: Key metrics across different subgroup configurations (continue)

| $n$ | $\ell$ | $n_1$ | $p_1$ | $\lceil \log p_1 \rceil$ | $\lceil \log p_1 - 1 \rceil$ | $R$ | $C_T$ (%) | $C_u$ (%) |
|-----|--------|-------|-------|--------------------------|------------------------------|-----|-----------|-----------|
| 50 | 1 | 50 | 51 | 6 | 5 | 60 | 360 (-) | 360 (-) |
| 50 | 2 | 25 | 29 | 5 | 4 | 34 | 340 (5.6%) | 170 (52.8%) |
| 50 | 5 | 10 | 11 | 4 | 3 | 16 | 320 (11.1%) | 64 (82.2%) |
| 50 | 10 | 5 | 7 | 3 | 2 | 8 | 240 (33.3%) | 24 (93.3%) |
| 60 | 1 | 60 | 61 | 6 | 5 | 72 | 432 (-) | 432 (-) |
| 60 | 2 | 30 | 31 | 5 | 4 | 38 | 380 (12.0%) | 190 (56.0%) |
| 60 | 3 | 20 | 23 | 5 | 3 | 32 | 480 (-11.1%) | 160 (63.0%) |
| 60 | 5 | 12 | 13 | 4 | 3 | 18 | 360 (16.7%) | 72 (83.3%) |
| 60 | 6 | 10 | 11 | 4 | 2 | 16 | 384 (11.1%) | 64 (85.2%) |
| 60 | 10 | 6 | 7 | 3 | 2 | 10 | 300 (30.6%) | 30 (93.1%) |
| 60 | 12 | 5 | 7 | 3 | 2 | 8 | 288 (33.3%) | 24 (94.4%) |
| 60 | 20 | 3 | 5 | 3 | 2 | 4 | 240 (44.4%) | 12 (97.2%) |
| 70 | 1 | 70 | 71 | 7 | 6 | 84 | 588 (-) | 588 (-) |
| 70 | 2 | 35 | 37 | 6 | 5 | 44 | 528 (10.2%) | 264 (55.1%) |
| 70 | 5 | 14 | 17 | 5 | 4 | 22 | 550 (6.5%) | 110 (81.3%) |
| 70 | 7 | 10 | 11 | 4 | 3 | 16 | 448 (23.8%) | 64 (89.1%) |
| 70 | 10 | 7 | 11 | 4 | 3 | 14 | 560 (4.8%) | 56 (90.5%) |
| 70 | 14 | 5 | 7 | 3 | 3 | 8 | 336 (42.9%) | 24 (95.9%) |
| 80 | 1 | 80 | 81 | 7 | 6 | 92 | 644 (-) | 644 (-) |
| 80 | 2 | 40 | 41 | 6 | 5 | 48 | 576 (10.6%) | 288 (55.3%) |
| 80 | 4 | 20 | 23 | 5 | 4 | 32 | 640 (0.6%) | 160 (75.2%) |
| 80 | 5 | 16 | 17 | 5 | 4 | 20 | 500 (22.4%) | 100 (84.5%) |
| 80 | 8 | 10 | 11 | 4 | 3 | 16 | 512 (20.6%) | 64 (90.1%) |
| 80 | 10 | 8 | 11 | 4 | 3 | 14 | 560 (13.0%) | 56 (91.3%) |
| 80 | 16 | 5 | 7 | 3 | 2 | 8 | 384 (40.4%) | 24 (96.3%) |
| 80 | 20 | 4 | 5 | 3 | 2 | 6 | 360 (44.1%) | 18 (97.2%) |
| 90 | 1 | 90 | 91 | 7 | 6 | 104 | 728 (-) | 728 (-) |
| 90 | 2 | 45 | 47 | 6 | 5 | 54 | 648 (11.0%) | 324 (55.5%) |
| 90 | 3 | 30 | 31 | 5 | 4 | 38 | 570 (21.7%) | 190 (73.9%) |
| 90 | 5 | 18 | 19 | 5 | 4 | 26 | 650 (10.7%) | 130 (82.1%) |
| 90 | 6 | 15 | 17 | 5 | 4 | 18 | 540 (25.8%) | 90 (87.6%) |
| 90 | 9 | 10 | 11 | 4 | 3 | 16 | 576 (20.9%) | 64 (91.2%) |
| 90 | 10 | 9 | 11 | 4 | 3 | 14 | 560 (23.1%) | 56 (92.3%) |
| 90 | 15 | 6 | 7 | 3 | 2 | 10 | 450 (38.2%) | 30 (95.9%) |
| 90 | 18 | 5 | 7 | 3 | 2 | 8 | 432 (40.7%) | 24 (96.7%) |
| 90 | 30 | 3 | 5 | 3 | 2 | 4 | 360 (50.5%) | 12 (98.4%) |
| 100 | 1 | 100 | 101 | 7 | 6 | 114 | 798 (-) | 798 (-) |
| 100 | 2 | 50 | 51 | 6 | 5 | 60 | 720 (9.8%) | 360 (54.9%) |
| 100 | 4 | 25 | 29 | 5 | 4 | 34 | 680 (14.8%) | 170 (78.7%) |
| 100 | 5 | 20 | 23 | 5 | 4 | 32 | 800 (-0.3%) | 160 (79.9%) |
| 100 | 10 | 10 | 11 | 4 | 3 | 16 | 640 (19.8%) | 64 (92.0%) |
| 100 | 20 | 5 | 7 | 3 | 2 | 8 | 480 (39.9%) | 24 (97.0%) |
| 100 | 25 | 4 | 5 | 3 | 2 | 6 | 450 (43.6%) | 18 (97.7%) |

