# OpenReview forum: "Hi-SAFE: Hierarchical Secure Aggregation for Lightweight Federated Learning"
_ICLR.cc/2026/Conference — ICLR 2026 Conference Withdrawn Submission_

### Official Review · Reviewer_95uL · 2025-10-28

**Soundness:** 2
**Presentation:** 3
**Contribution:** 3
**Rating:** 2
**Confidence:** 4

**Summary:**

This paper builds upon the foundation of signSGD with majority vote and designs a new hierarchical secure aggregation protocol to ensure security and prevent information leakage. By subgrouping clients, the framework enables communication efficiency and security.

**Strengths:**

* Distributed machine learning is always concerned with communication efficiency and privacy leakage;
* The idea of using secure aggregation to enhance the privacy guarantees of signsgd is innovative.

**Weaknesses:**

* The abstract can be polished up. For example, the authors talk about constant multiplicative depth and bounded per-user complexity in lines 22- 24. Yet, neither are these standard, nor the definitions are given.
* Does $Enc(\mathbf{x}_i)$ remain to be sign bits after using Eq.(3) in line 223? Since the operation is concluded by a $\mod p$, it is temptive to think the produced result is a quantized level.
* Theorem 1 requires clarification.
  * First, the assumptions are not included in the theorem statement. Upon checking the appendix, it adopts almost the same set of assumptions as (Bernstein et al. 2018a). However, it is an open question whether is possible to attain a mini-batch size as large as $m_k = K$, where $K$ is the number of total rounds.
  * Second, probability $q$ of a correct majority vote is not stated in Theorem 1 or any auxiliary lemmas.
* The experiements are quite limited in that only non-subgrouping and subgrouping are compared.
  * More baselines should be included. Right now, the results are more like ablation studies.
  * Should also include the probability of a correct majority vote.
  * Concrete privacy leakage measurements should also be included.

**Questions:**

* Can the authors make another round of proofread to their abstract to improve quality?
* Can the authors clarify the number of bits used by Eq. (3)?
* Can the authors double check their proof and clarify their theorem statement? More importantly, they would want to address the scaling of their mini-batch size.
* Can the authors include the auxiliary results for the probability $q$ of a correct majority vote?
* Can the authors provide more numerical results?

---

### Official Review · Reviewer_t8hX · 2025-10-31

**Soundness:** 3
**Presentation:** 3
**Contribution:** 2
**Rating:** 4
**Confidence:** 4

**Summary:**

The paper proposes Hi-SAFE, a secure aggregation framework that integrates SignSGD with a polynomial formulation of majority voting based on Fermat’s Little Theorem. To reduce the computation and communication cost, the author consider to divide the clients into small group, that would reduce the prime $p$ and the degree of polynomial fuction $F(x)$.  While the motivation—achieving lightweight secure aggregation for sign-based privacy-preserving federated learning (PPFL)—is relevant, several conceptual and technical aspects require further clarification.

**Strengths:**

1. A novel approach involves constructing polynomial functions based on Fermat's Little Theorem to evaluate the secure aggregation results of SIGNSGD.

2. The formal analysis of the problem and the description of the algorithm are well-structured and clear at a glance.

3. The appendix provides a wealth of examples to help readers better understand the paper.

**Weaknesses:**

1.Limited novelty in applying SignSGD to PPFL. The paper positions the integration of SignSGD and secure aggregation as novel. However, there already exists a substantial body of work that leverages quantization techniques similar to SignSGD for communication-efficient or privacy-preserving federated learning, including but not limited to [1], [2] and [3]. The manuscript does not sufficiently differentiate Hi-SAFE from these prior efforts.

2.Misleading claim of communication efficiency. The proposed protocol operates over a prime field $\mathbb{F}_{p}$, and all operations (including Beaver triples) depend on the modulus $p$, which directly determines the communication volume per encrypted round. In such cases, the theoretical 1-bit advantage of SignSGD becomes negligible.

Moreover, the polynomial-based method adopted in this work introduces an additional $O(n^{2})$ computational complexity, which leads to significant computational overhead within the finite field and consequently causes high runtime latency. While the proposed subgrouping strategy is conceptually appealing for reducing the computational and communication overhead of secure aggregation, its practical feasibility remains questionable. The theoretical convergence analysis relies on independence assumptions among subgroups, which may not hold in realistic Non-IID federated settings where client gradients are correlated, potentially leading to biased global updates.

Furthermore, the paper lacks empirical runtime or bandwidth measurements and does not compare the subgrouping approach against existing lightweight secure aggregation baselines such as LightSecAgg (So et al., 2022) or FedLSC (Joo et al., 2025). As a result, the claimed efficiency and scalability benefits of the subgrouping mechanism remain largely unsubstantiated.


[1] Dong, Ye, et al. "FLOD: Oblivious defender for private Byzantine-robust federated learning with dishonest-majority." European Symposium on Research in Computer Security. Cham: Springer International Publishing, 2021.

[2] Miao, Yinbin, et al. "Privacy-preserving Byzantine-robust federated learning via blockchain systems." IEEE Transactions on Information Forensics and Security 17 (2022): 2848-2861.

[3] So, Jinhyun, Başak Güler, and A. Salman Avestimehr. "Byzantine-resilient secure federated learning." IEEE Journal on Selected Areas in Communications 39.7 (2020): 2168-2181.

**Questions:**

see the above

---

### Official Review · Reviewer_pGxy · 2025-10-31

**Soundness:** 2
**Presentation:** 3
**Contribution:** 2
**Rating:** 2
**Confidence:** 4

**Summary:**

This paper proposes Hi-SAFE, a cryptographic secure-aggregation framework tailored to SIGNSGD-MV in federated learning.

**Strengths:**

The paper targets a timely and important gap: combining 1-bit communication efficiency with end-to-end cryptographic privacy for sign-based FL.

**Weaknesses:**

Key technical claims require clarification or stronger evidence.

(i)  “majority of subgroup majorities” is not generally equivalent to the global majority unless nontrivial conditions hold (balanced subgroup sizes, odd sizes, consistent tie-breaking, no near-ties);

(ii) With (p!>!n), the indicator term (1-(x-m)^{p-1}) implies (\deg(F)=p-1=\Theta(n)); it is unclear how the protocol attains constant multiplicative depth and ≤6 secure multiplications per user. The recurrence with (v_k) suggests at best (O(\log n)) multiplicative depth;

(iii) SIGNSGD-MV is coordinate-wise; evaluating (F(\cdot)) per parameter dimension (d) multiplies MPC cost and bandwidth by (d). The paper should quantify per-round cost vs. (d) for modern models, include memory for precomputed coefficients, and show that benefits persist beyond small CNNs.

(iv) Revealing only the final majority still leaks information across rounds (e.g., inferring cohort drift or individual influence given known local signs). Provide a concrete leakage analysis or DP accounting, not just semi-honest security prose.

(v) Who generates triples in FL without a trusted dealer? What is the offline bandwidth/latency on constrained devices, and how are dropouts/stragglers handled without breaking triple consistency? Include protocols for client churn, subgroup reconfiguration, and replay/abort safety.

(vi) Sign-based methods can struggle under non-IID and are sensitive to adversarial sign flips; despite a semi-honest model, poisoning and Sybil resilience matter in practice.

**Questions:**

1. Please state and prove precise conditions under which “majority of subgroup majorities” equals the true global majority (e.g., equal subgroup sizes, odd n1, strict ties handled consistently).

2. Algorithm 3 (Step 9) says the server “reconstructs F(xj) for each subgroup,” which appears to reveal subgroup majorities (s_j) to the server—contradicting the claim that only the final (s) is revealed.

3. With (p>n), the indicator (1-(x-m)^{p-1}) implies (\deg(F)=p-1=\Theta(n)). How do you attain “constant depth (~2)” and “≤6 secure multiplications per user”?

4. Figure 3 asserts a constant bound (≤6) independent of (n). Please provide a formal derivation tying this bound to (n_1), the chosen (p_1), and the exponentiation schedule, and specify for which ((n,\ell)) this bound holds.

5. You define latency as (\lceil \log(p_1-1)\rceil) “serial depth for Beaver triple multiplication.” Why does depth scale with (\log(p_1-1)) rather than with the number of multiplication gates in your concrete circuit for evaluating (F)?

6. SIGNSGD-MV is coordinate-wise. What is the total online communication and triple consumption for a model with (d) parameters?

7. Specify the encoding, how negative values are represented, and why the sign polynomial is correct modulo (p) for all coordinates.

8. Provide ablations showing how 1-bit vs 2-bit tie-breaking affects optimization dynamics (stability, plateaus) across datasets and non-IID regimes.

9. Theorem 1 assumes each subgroup outputs the correct majority with probability (q>0.5), independent across subgroups. In practice, subgroup outcomes can be correlated (due to shared data heterogeneity and client selection).

10. Theorem 1 uses (\eta=1/(K|\vec L|_1)) and (m_k=K). This is unusual for practical FL.

11. The bound contains (c',\ell^{-1/4}\exp(-\ell \alpha_q^2/4)). Define (c') precisely, show how it depends on aggregation perturbations, and correct any typographical ambiguities (is it (\exp{-\ell \alpha_q^2/4})?).

12. Theorem 2 assumes at least one honest user per subgroup. What if a subgroup is fully corrupted (or all but one user drop out)? Also, does your semi-honest adversary include the server colluding with a subset of clients?

13. Give a detailed real/ideal proof sketch: what are the simulator’s views in both stages (subgroup and inter-group), how are messages simulated from only (({x_{i,j}}_{i\in C}, s)), and where do Beaver triple masks guarantee indistinguishability?

14. Who generates triples (trusted dealer, OT-based COT/VOLE like MASCOT, Silent OT/Ferret, or a pair of non-colluding servers)?

15. Even if privacy is semi-honest, correctness can be attacked by sign-flip or Sybil clients.

16. The statement that leakage arises only when all inputs are identical (prob. (2^{-(n-1)})) is too strong.

17. How does the protocol handle client dropouts mid-round without revealing masked differences or invalidating shares?

18. In realistic FL, the number of active clients varies per round. How do you pick (p_1>n_1) and (p_2>\ell) when (n_1,\ell) change?

19. Provide a systematic sweep over (\ell) showing accuracy/convergence, per-user bytes, triples, and round latency, including non-IID partitions (Dirichlet (\alpha)) and heterogeneous compute/bandwidth.

20. Several critical statements (“constant depth,” “≤6 multiplications,” “often depth 2,” “server-side intra-subgroup incurs no uplink”) need explicit references to equations/lemmas in Appendices E–I.

21. If subgroup ties occur often (e.g., highly skewed or tiny (n_1)), what is the recommended policy (2-bit ties, random tie-break, or adaptive subgroup resizing)?

22. Current experiments (MNIST/FMNIST/CIFAR-10 on 2×3090) are not representative of edge deployment.

23. Unify scalar vs. vector notation (define (x) per coordinate), define (N_t), and ensure equations (1)–(3), (5)–(8) align with the algorithms.

---

### Official Review · Reviewer_opuM · 2025-11-03

**Soundness:** 3
**Presentation:** 3
**Contribution:** 3
**Rating:** 6
**Confidence:** 3

**Summary:**

The paper proposes Hi-SAFE, a secure aggregation framework tailored to SIGNSGD-MV. The key idea is to express the coordinate-wise majority vote as a low-degree polynomial over a prime field using Fermat’s Little Theorem (FLT), then evaluate it with additive secret sharing and Beaver triples so that only the final vote is revealed. To keep multiplication depth and per-user cost constant, the system introduces hierarchical subgrouping (intra-group secure vote, then inter-group aggregation). This paper includes theoretical analysis for convergence and security analysis, which is standard.  Experiments on MNIST/FMNIST/CIFAR-10 report large per-user communication reductions with comparable accuracy.

**Strengths:**

- Using FLT-based indicator for secure aggregation of sign-sgd is smart and the construction is well stated.
- The subgrouping design keeps multiplicative depth constant, independent of the number of users, $n$.
- Protocol descriptions are concrete and well presented.
- Theoretical convergence analysis and security proof are included.

**Weaknesses:**

- This paper has no clear experimental or analytical comparison against existing approaches that approximate or compute the sign function using polynomial representations combined with secure computation techniques (e.g., homomorphic encryption, MPC-based sign-SGD). There is no baseline MPC-based sign-SGD in
- While experiment section provides communication reduction from the sub-group strategy (compared to the original version of the proposed scheme), asymptotic communcation and computation analysis is missing.

**Questions:**

- Several prior MPC and HE studies have explored polynomial approximations to sign and ReLU for privacy-preserving learning. Could the authors clarify how Hi-SAFE’s polynomial design and Beaver-based evaluation differ fundamentally from these works? If possible, could experimental comparison in terms of model accuracy, latency (total running time), and communication overhead be provided?
- As described in the step 9 of algorithm 3, does the server reconstruct $s_j = F(x_j)$? If yes, please explain why this does not leak the lone honest user’s sign when a subgroup has $n_1 -1$ colluding users whose partial sum could be 0 (then the sign of the honest user reveals). If no, please clarify step 9 in algorithm 3.
- What is the communication and computation overhead as function of important system parameters (such as $n, l, p_1, ...$)?

---

### Note · Authors · 2025-11-18

I have read and agree with the venue's withdrawal policy on behalf of myself and my co-authors.